



# Establishing Long-term Measurements of Halocarbons at Taunus Observatory

Tanja J. Schuck[1], Fides Lefrancois[1], Franziska Gallmann[1], Danrong Wang[1], Markus Jesswein[1],
Jesica Hoker[1], Harald Bönisch[1,2], and Andreas Engel[1]

[1]Goethe University Frankfurt, Germany
[2] now at: Karlsruhe Institute of Technology, Germany

*Correspondence to:* T. J. Schuck (schuck@iau.uni-frankfurt.de)

**Abstract.**

In late 2013, a whole air flask collection program started at the Taunus Observatory (TO) in central Germany. Being a rural site in close vicinity to the densely populated Rhein-Main area, Taunus Observatory allows to assess local and regional emissions. Owed to its altitude of 825 m, the site also regularly experiences background conditions, especially when air masses

approach from north-westerly directions. With a large footprint area mainly covering central Europe north of the Alps, halocarbon measurements at the site have the potential to improve the data base for estimation of regional and total European halogenated greenhouse gas emissions. Flask samples are collected weekly for offline analysis using a GC-MS system employing a quadrupole as well as a time-of-flight mass spectrometer. As background reference, additional samples are collected approximately bi-weekly at the Mace Head Atmospheric Research Station (MHD) when air masses approach from the site's

clean air sector. Thus the TO time series can be linked to the in-situ AGAGE measurements and the NOAA flask sampling program at MHD. An iterative baseline identification procedure separates polluted samples from baseline data. While there is good agreement of baseline mixing ratios between TO and MHD, with a larger variability of mixing ratios at the continental site, measurements at TO are regularly influenced by elevated halocarbon mixing ratios. Here, first time series are presented for CFC-11, CFC-12, HCFC-22, HFC-134a, HFC-227ea, HFC-245fa, and dichloromethane. While atmospheric mixing ratios

of the CFCs decrease, they increase for the HCFC and the HFCs. Small unexpected differences between CFC-11 and CFC-12 are found with regard to the occurrence of high mixing ratio events and seasonality, although production and use of both compounds are strictly regulated by the Montreal Protocol, and therefore a similar decrease of atmospheric mixing ratios should occur. Dichloromethane, a solvent about which recently concerns have risen regarding its growing influence on stratospheric ozone depletion, does not show a significant trend with regard to both, baseline mixing ratios and the occurrence of pollution

events at Taunus Observatory for the time period covered, indicating stable emissions in the regions that influence the site. An analysis of HYSPLIT trajectories reveals differences in halocarbon mixing ranges depending on air mass origin.

## 1  Introduction

Halogenated trace gases play an important role in atmospheric chemistry, they contribute to the depletion of stratospheric ozone and directly or indirectly to the radiative forcing of the atmosphere (Carpenter et al., 2014; IPCC, 2013). Many of them

do not have natural sources, but are purely anthropogenic. Their use includes various applications such as refrigeration, air conditioning, fire extinguishers or foam blowing. As a consequence of the regulation of their production and use in the Montreal Protocol and its amendments, mixing ratios of anthropogenic halocarbons in the atmosphere exhibit strong trends. Mixing ratios of chlorofluorocarbons (CFC) and compounds such as long-lived chlorinated solvents (e.g. $CCl_4$) have started to decrease since





the 1990s. While mixing ratios still decrease, recently an increase in CFC-11 emissions was observed within the US National Oceanic and Atmospheric Administration (NOAA) network, pointing to new production of this compound (Montzka et al., 2018) and showing the need of continued monitoring.

The Montreal Protocol also regulates production and use of hydrochlorofluorocarbons (HCFCs), the first-generation replace-
ment substances of CFCs, but phase-out is not yet fully accomplished. Owing to their long lifetimes many hydrochlorofluoro-carbons still accumulate in the atmosphere or are just about to level off. Use of long-lived compounds from the next generation replacements, hydrofluorocarbons (HFC), has only recently been included in the Montreal Protocol, and atmospheric mixing ratios of halocarbons from this group are currently still increasing (Simmonds et al., 2017; Montzka et al., 2015). As fur-ther replacement substances new short-lived unsaturated HFCs (also called HFOs, hydrofluoroolefins) have already reached
detectable mixing ratios in the atmosphere (Vollmer et al., 2015).

For many atmospheric trace gases, including halogenated compounds, regular ground-based measurements at fixed sites provide the main data basis to study changes in atmospheric composition. Measurements often take place at remote sites such as mountain tops or coastal locations, far from emission sources and representative of a large catchment area. This reduces the influence of local or regional emissions and allows to study changes in the composition of the atmospheric background. Sites
at which regular measurements of halogenated trace gases are currently performed in Europe are Jungfraujoch (3580 m.a.s.l., Switzerland), Monte Cimone (2165 m.a.s.l., Italy), Zeppelin Observatory (490 m.a.s.l, Norway), and Mace Head (25 m.a.s.l., Ireland), all part of or affiliated with the network of the Advanced Global Atmospheric Gases Experiment (AGAGE) (Prinn et al., 2018). Of these, Jungfraujoch is characterised by mainly free tropospheric air masses, but the site also experiences events of transport from the boundary layer. Also Monte Cimone does regularly experience regional pollution events. A comparison
of 34 European observation sites classified Mace Head and Jungfraujoch as remote sites and Monte Cimone as a site weakly influenced by emissions (Henne et al., 2010). Data from the high-latitude Zeppelin Observatory do not seem to strongly constrain European emissions (Maione et al., 2014).

Several recent studies have combined station measurements of halocarbons and atmospheric transport models to inversely estimate emissions on different geographical scales (e.g. Keller et al., 2012; Maione et al., 2014; O'Doherty et al., 2014;
Simmonds et al., 2016; Brunner et al., 2017; Hu et al., 2017). This approach has the potential to improve existing emission inventories and can also serve for the verification of emissions reported to the United Nations Framework Convention on Climate Change (UNFCCC) on the European level. Comparing four different model approaches for estimates of national emissions, Brunner et al. (2017) found large differences of a factor of 2.4 between reported emissions of HFC-125a and the model median, suggesting that bottom-up estimates for emissions from Germany of this compound and also for HFC-134a are
too low. This result agrees with a previous study, which suggested significantly underestimated emissions from Germany also for HFC-143a (Lunt et al., 2015). These studies show the need for additional observations in Germany.

Inversion-based emission estimates rely on high-quality observations of trace gas mixing ratios. Thus, they are currently limited by the sparse distribution of long-term observational sites. For long-lived gases, such as CFCs and HCFCs, the present network of a small number of representative background stations is sufficient to estimate emissions on the global or hemispheric
scale. However, with more shorter-lived species coming into the scientific focus and emission estimates aiming at the national or regional scale, a denser network is required (Brunner et al., 2017; Villani et al., 2010). In Europe, the surface sensitivity of the current observational network, and thus the ability of the observations to constrain the modelled emission estimates, decreases over the northern parts of Germany, the Benelux region, and Eastern Europe (Brunner et al., 2017; Maione et al.,


2014). An observation site in central Germany could improve this situation by enhancing the sensitivity to emissions from Germany and potentially also from the Benelux region and France, as westerly winds commonly occur (cf. Figure 1).

To assess regional emissions of halocarbons, in particular emissions from Germany, a flask sampling program was started at Taunus Observatory in Germany in late 2013. Here we report on the first years of data for selected halogenated compounds.

The measurements include a large suite of more than 40 target species of chlorine-, bromine- and iodine-containing gases. In addition, non-target information is available from the time-of-flight mass spectrometer. More than 50 compounds have been identified in the mass spectra from this instrument.

Here, we focus on chlorinated gases, among them chlorofluorocarbons and hydrochlorofluorocarbons, of which exemplary CFC-11 ($CFCl_3$), CFC-12 ($CF_2Cl_22$) and HCFC-22 ($CHClF_2$) are shown, being the most abundant CFCs and the most abun-

dant HCFC in the atmosphere. As examples of long-lived hydrofluorocarbons HFC-134a ($CH_2FCF_3$), the most abundant compound of this group, and HFC-245fa ($CF_3CHCF_2H$) and HFC-227ea ($CF_3CHFCF_3$) are discussed. Atmospheric mixing ratios of these compounds are still growing. In addition, measurement results for dichloromethane ($CH_2Cl_2$) are shown because this solvent was recently discussed in the literature, and concerns were risen that its use and in consequence its emissions might increase with further economic growth in Asia (Hossaini et al., 2017; Oram et al., 2017). Then dichloromethane could become

a more relevant source of stratospheric chlorine in the future possibly enhancing stratospheric ozone depletion.

## 2 Measurements

### 2.1 Sample Collection at Taunus Observatory

Taunus Observatory is located at 50.22 ° N, 8.44° E at 825 m altitude on top of Kleiner Feldberg in the Taunus mountain range. Northward, the area is dominated by forest, which needs to be taken into account for biogenic substances, and agriculture.

Approx. 20 km south-east of the site is the city of Frankfurt (Main) in the centre of the Rhein-Main area with several industrial sources including chemical industry. Wind direction is predominantly from the west. Figure 1 shows the surface sensitivity of the site derived from bi-weekly dispersion model calculations (FLEXPART) for the year 2007. Measurements at TO are expected to provide additional constraints to emission estimates for south-west Germany, France and the Benelux region, but occasionally also air masses from north-westerly or easterly directions are encountered. During night time inversion, the site is

usually located above the top of the planetary boundary layer. This may also be the case during daytime, in particular in winter when inversions can persist over several days. This could be a challenge for models trying to capture the observed variability because of the limitations of the spatial resolution of modelled transport.

Regular sample collection at Taunus Observatory started in October 2013 and is ongoing. Samples are collected during daytime on a weekly basis, irrespective of meteorological parameters such as wind direction or wind speed. Sampling happens

through stainless steel tubing at an intake height of approx. 8 m above ground. For quality assurance, two stainless steel canisters are pressurized in parallel up to 2.5 bar after flushing with ambient air for 15 min using a metal bellows pump.

To assess European background mixing ratios and also to link the TO time series to existing long-term international programs, flask sampling collection is also performed regularly at the Mace Head Research Station (MHD) in Ireland. Here regular in-situ measurements of the AGAGE program and sample collection for the NOAA network take place. Since March

2014 additional samples are collected approximately bi-weekly during periods when air masses approach from the clean air sector at Mace Head. The data thus represent the expected baseline for atmospheric mixing ratios at Taunus Observatory. To



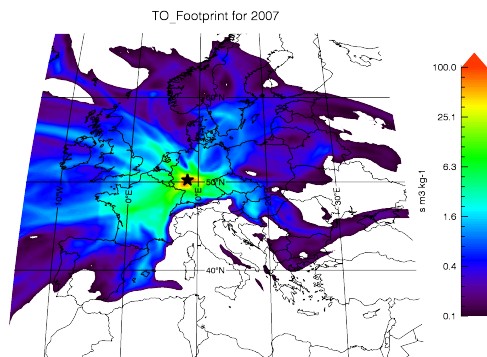

**Figure 1.** Contour plot of source sensitivity of Taunus Observatory derived from bi-weekly particle dispersion calculation for the year 2007. Taunus Observatory is located at 50.22° N, 8.44° E. Data courtesy of D. Brunner.

facilitate comparison with data from the NOAA network, sample collection usually takes place directly after sample collection for NOAA, using the same equipment. Samples are analysed after shipment to Frankfurt.

## 2.2 Instrumentation

Measurements of halocarbons at Taunus Observatory are based on gas chromatograph mass spectrometry (GC/MS) following
cryogenic enrichment of samples. Technical details of the employed setup are given in Hoker et al. (2015) and will only briefly be reviewed here. Halocarbon mixing ratios in the atmosphere range from a few ppt up to a few hundred ppt. Therefore measurements require pre-concentration of the sample air prior to gas chromatographic separation and detection. Pre-concentration is achieved by passing the sample flow of 150 ml/min through an adsorption material (HayeSep D) at -80 °C. For cooling, the 1/16" stainless steel sample loop is placed inside an aluminium block which is kept at -80 °C by a Stirling cooler (Global Cool-
ing, M150). The enriched sample volume is determined by monitoring the pressure inside a 2 x 2 l reference volume which gets evacuated prior to sample enrichment. After enrichment of 1 l of sample volume, the sample loop is heated to approx. 200 °C for 4 min while the carrier gas flow is directed through it (purified Helium 6.0, Purification System: Vici Valco HP2). Prior to enrichment, the sample air is dried by passing a heated (80 °C) tube filled with magnesium perchlorate $Mg(ClO_4)_2$. Mixing ratios are therefore reported as dry mole fractions.

A gas chromatograph (Agilent 7890A) is used with a 7.5 m pre-column and a 22.5 m main column (both GasPro PLOT, inner diameter 0.32 mm). The temperature program of the GC starts at 50 °C kept for 2 min after which the oven is heated to 95 °C at a rate of 15 °C/min. Then it is heated to 135 °C at 10 °C/min, and finally to 200 °C at a rate of 22 °C/min. This temperature is kept for another 2.95 min. The complete runtime adds up to 17.95 min. Backward flushing of the pre-column is started after 12.6 min to avoid contamination of the subsequent chromatographic run with high-boiling substances.

Behind the main column the gas flow is split (ratio approx. 0.6:0.4) into two fused silica transfer lines connected to a quadrupole mass spectrometer (Agilent 5975C) and a time-of-flight mass spectrometer (Markes Bench TOF-dx E-24). The quadrupole-MS is operated in selected ion monitoring (SIM) mode, scanning pre-selected masses at a given retention time. The time-of-flight-MS scans the mass range from 45–500 amu. Ionisation for both instruments is via electron impact at 70 eV.



Extending the setup described by Hoker et al. (2015), the system has been automated for unattended operation of up to ten individual sample canisters in one sequence. This has been achieved with pressure-operated on/off valves (Vici Valco AS-FVO2HT3) for stream selection (helium, sample, standard) and a 10-port multi-position valve (Vici Valco EMT2SD10MWE) for sample selection. All valves are heated and kept at temperatures around 80 °C.

Each air sample is analysed twice, each double measurement being bracketed by a single measurement of a whole air standard which was cryogenically filled in December 2007 at Jungfraujoch, Switzerland. Mixing ratios of this working standard have been calibrated against an AGAGE gas standard. All data are reported on AGAGE scales as listed in table 1. A full measurement series also includes a blank measurement of the purified helium used as carrier gas, a vacuum blank and a measurement of a target standard.

Chromatographic peaks are integrated with a custom designed software written in the programming language IDL. The peak fitting algorithm applies Gaussian fits with a constant or linear baseline. Noise calculation is performed on baseline sections close to peak retention times by determining the threefold standard deviation of the residuals between baseline data points and a second order polynomial fit. Peaks with a signal-to-noise ratio below 1.5 are rejected. The integrated detector signal is normalised to the exact enriched sample volume, determined by a pressure measurement. To account for detector drift during

measurement series, the calibration measurements bracketing the sample pairs are interpolated linearly. The relative response for each sample is calculated as the ratio between sample and corresponding interpolated calibration point.

### 2.3   Data Quality and Long-term Stability of Measurements

To ensure a high-quality dataset, automated procedures filter the data based on instrumental precision. The precision was determined for each substance and individually for the two mass spectrometers based on two sequences of 20 measurements of

the Jungfraujoch working standard as described in Hoker et al. (2015). After changes were made to the enrichment unit in 2016, the reproducibility experiment was repeated with no significant difference from the previous results. In a second assessment of reproducibility the instrument precision was determined using another working standard pressurized at Taunus Observatory in 2015. In this experiment, the TO working standard was analysed 13 times in a measurement sequence following the same procedure as regular air samples, and this was repeated on three different days. Instrument precision was calculated as the

standard deviation of these measurements after application of the drift correction. Instrumental precisions derived from the second working standard in this way agreed with values from Hoker et al. (2015), thus for consistency the latter are used in the following. Table 1 lists precisions of the substances presented here.

Precision values range for the quadrupole MS from 0.14 % (CFC-11) to 9.2 % (HFC-245fa). For the time-of-flight-MS relative precisions range from 0.20 %(CFC-11) to 9.4 % (HCFC-225cb). For most substances the quadrupole-MS yields a

slightly better precision, which may partly be due to the split ratio of the gas flow. Therefore, and because TOF data do not cover the time after September 2017, quadrupole data are shown if not mentioned otherwise.

Based on the instrumental precision, two types of filter routines are applied after integration of the chromatograms and calculation of drift corrected relative responses:

i) Precision criterion: for the two analyses performed for each sample canister, the standard deviation of the two resulting

values of the relative response are calculated and compared with the instrumental precision for each substance. If the standard deviation of the double analysis exceeds three times the system precision, the sample analysis is rejected.

ii) Overlap criterion: in addition to double analysis of each sample, canisters are collected in pairs. For each pair it is checked, whether the results agree within $2\sigma$, $\sigma$ being the standard deviation of the double analysis of each canister.



**Table 1.** System precision (prc) in % for selected substances for detection with the quadrupole (QP) and the time-of-flight mass spectrometer (TOF). The value of the system precision represents the best repeatability for the system as deduced from dedicated measurements. Precision for a particular measurement day can be different. Mixing ratios are reported as dry mixing ratios on AGAGE scales as listed in the first column. Columns labelled standard-1 and standard-2 contain long-term stability deduced from measurements of two primary standards. Numbers in brackets give the number of measurements available for the respective instrument. Listed are standard deviations of all precision filtered data.

| compound | scale | QP prc | QP standard-1 (25) | QP standard-2 (56) | TOF prc | TOF standard-1 (22) | TOF standard-2 (42) |
|---|---|---|---|---|---|---|---|
| CFC-11 | SIO-05 | 0.14 % | 0.37 % | 0.26 % | 0.20 % | 0.28 % | 0.61 % |
| CFC-12 | SIO-05 | 0.32 % | 0.37 % | 0.32 % | 0.29 % | 0.53 % | 0.48 % |
| HCFC-22 | SIO-05 | 0.36 % | 0.56 % | 0.90 % | 0.82 % | 0.40 % | 0.96 % |
| HFC-134a | SIO-05 | 0.47 % | 1.7 % | 0.51 % | 0.41 % | 0.78 % | 1.4 % |
| HFC-227ea | SIO-14 | 0.22 % | — % | 12 % | 7.1 % | — | 6.9 % |
| HFC-245fa | SIO-14 | 9.2 % | — % | 4.2 % | 1.6 % | — | 6.7 % |
| Dichloromethane | SIO-14 | 0.48 % | 1.8 % | 2.5 % | 1.2 % | 0.64 % | 3.7 % |

Only if both criteria are fulfilled, the data are included in the final time series. The mixing ratio is then calculated from the mean relative response of the detector for the sample pair. If for a pair sufficient overlap was diagnosed but only one canister meets the precision criterion, the precision rejected sample is excluded and the mixing ratio is calculated for the remaining canister only. Data which do not meet both, the precision criteria of threefold precision and the $2\sigma$ overlap criterion for double

samples, are excluded from further analysis.

For each sample measurement day, an average daily value of the system precision is calculated from the standard deviation of the double analyses that have met the precision limit. The relative error of each final mixing ratio is reported as either the daily precision of the day when the canisters were analysed or the instrumental precision derived from the dedicated experiment, whichever is larger.

To monitor long-term stability of the GC-MS system, a primary standard is measured as target at least once per month. This is usually done as part of a regular sample measurement routine, measuring the target standard relative to the working standard. Since the working standard has been calibrated versus the two primary standards, this procedure checks for relative drifts of the standards. The target standard is treated as an air sample in this procedure, and data are filtered for data precision as described above for air samples. Two different target standards were used for this, standard-1 being measured regularly

from October 2013 through October 2014, standard-2 since then. Individual measurements of standard-1 were also performed in 2017. Slopes of the obtained target time series agree with 0 confirming no relative drift of the primary and the working standards. Table 1 lists the corresponding standard deviations for both mass spectrometers and both standards.

While the system precisions in Table 1 reflect the repeatability of measurements on short time scales (i. e. hours), these target measurements assess long-term stability of the GC-MS system and the calibration standard. Ideally, both standard deviations

should be comparable, but system precision represents a lower limit to the variability of standard measurements on the time scales of years. Standard deviations of the target measurements in Table 1 are comparable with system precisions for most substances but deviate for the three HFCs and for dichloromethane. In standard-2, mixing ratios of HFC-134a, HFC-245fa and dichloromethane are markedly below current atmospheric mixing ratios and below the mixing ratios of the working standard





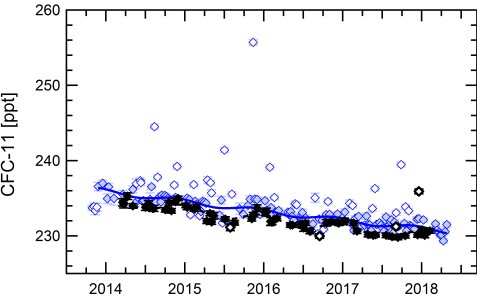 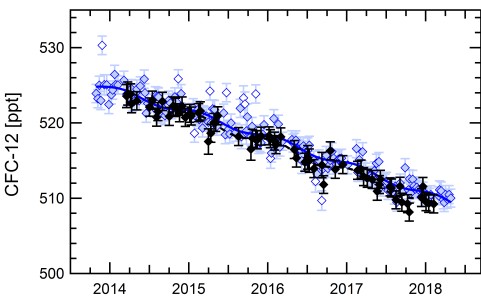

**Figure 2.** Time series of CFC-11 (a) and CFC-12 (b) at Taunus Observatory (coloured symbols) and Mace Head (black symbols). The solid line represents a 2nd order polynomial harmonic fit to the TO baseline dataset, the dashed line the corresponding fit to the Mace Head data. Open symbols denote samples flagged as outliers.

which was used for calibration and to determine the system precisions given in Table 1. Thus, the signal-to-noise ratio of peaks gets smaller which can worsen repeatability (cf. e. g. Fig. 5 of (Obersteiner et al., 2016)). This could explain the higher variability in comparison to system precision. For standard-1 the number of high precision measurements of HFC-227ea and HFC-245fa was too small for statistical analysis.

The filter procedures outlined before yield a high quality data set. The filtering is only based on precision and data consistency but does not interpret measured mixing ratios. In a further step this data set is evaluated to distinguish between background measurements, i. e. baseline data, and outlier data points potentially influenced by regional emissions.

The baseline data are identified by fitting the following function:

$$\chi(t) = a + b \cdot t + c \cdot t^2 + d \cdot \sin(2\pi t) + e \cdot \cos(2\pi t). \tag{1}$$

Data outside a $2\sigma$-band around the residual mean are flagged as outliers. The remaining data are fitted again and data points which fall outside $2\sigma$ of the new residual are again flagged as outliers. This is iterated until the mean of the residual does not change by more than 10 % in the subsequent iteration. If in one step the standard deviation of the residual is smaller than the mean error of mixing ratios for a specific substance, the latter is used instead. This procedure was adopted similar to the AGAGE pollution identification algorithm (cf. (O'Doherty et al., 2009) and references therein). While it is expected that

outliers are mostly caused by pollution with mixing ratios above the baseline, outliers below the baseline can for example be due to a stratospheric influence when the aged stratospheric air contains lower mixing ratios or due to transport from lower latitudes for substances which exhibit a latitudinal gradient.

Application of the data quality filters and the outlier filter yield a quality assessed dataset separated into baseline data and outlier events. As an example Fig. 2 compares mixing ratios of CFC-12 and CFC-11 at Taunus Observatory and at Mace Head

Station at the west coast of Ireland. The Mace Head data serve as a reference for the data acquired at Taunus Observatory. Mace Head data are also quality filtered as explained above, including the outlier selection procedure. Because samples at Mace Head are collected when air is approaching from the clean wind sector of the site, the dataset is biased towards lower mixing ratios and the number of outliers is very small.





### 2.3.1 Comparison with Results from the NOAA Network

The Frankfurt GC-QP-MS system was characterised and used before for studies by e. g. Laube and Engel (2008); Brinckmann et al. (2012) and Hoker et al. (2015), and good agreement with the NOAA and AGAGE networks was achieved in the international comparison IHALACE (International Halocarbons in Air Comparison Experiment) (Hall et al., 2014). As mentioned before, sample collection at Mace Head is synchronized with sample collection for the NOAA network. Of the selected substances discussed here, NOAA GC/MS data are available from ftp://ftp.cmdl.noaa.gov/hats/ for CFC-11, HCFC-22, HFC-134a, HFC-227ea, and dichloromethane. NOAA data are updates to data included in Montzka et al. (2015); Hossaini et al. (2017); Montzka et al. (2018). Results from the Frankfurt GC/MS system are reported on AGAGE scales and data have not been corrected for scale differences. These are typically less than 3 % (Hall et al., 2014; Carpenter et al., 2014), therefore this does not have a major impact on the correlations which are shown in Fig. 3 for CFC-11, HCFC-22, HFC-134a, HFC-227ea, and dichloromethane. For CFC-12, NOAA GC/ECD data are available in high quality through early 2015. The short period of overlap between the two datasets does not allow a meaningful comparison.

Sampling for the NOAA network and for the dataset presented here is done sequentially. Mechanical connection of the samples and canister flushing amount to a time lag of typically 30–60 min. Although sampling is from the clean air sector, both data sets still contain some outliers with elevated mixing ratios. However, because sampling is not parallel but sequential, outliers in one data set that arise from atmospheric variability are in general not apparent in the other.

Correlation coefficients $r^2$ above 0.9 are obtained for all substances discussed here, except for CFC-11 ($r^2 = 0.81$). A special case is HFC-134a, for which good agreement with NOAA data is obtained with the quadrupole instrument, but data from the time-of-flight mass spectrometer deviate for mixing ratios above 90 ppt. A similar result is obtained when correlating data from both instruments at TO, pointing to a non-linearity of the time-of-flight mass spectrometer for HFC-134a. This is apparent from Fig. 4 showing mixing ratios of HFC-134a from the TOF-MS to deviate from the 1:1-line. The working standard used contains a HFC-134a mixing ratio of 53.25 ppt, significantly below current atmospheric values. Non-linear behaviour of the instrument was already determined for some substances by Hoker et al. (2015). Because of this known issue, only quadrupole data are shown for HFC-134a. Fig. 4 also shows the correlation of data from both instruments for HFC-245fa, for which a correlation slope of $0.97 \pm 0.04$ is obtained.

## 3 Results

### 3.1 Trends and Seasonality

Air sample collection at Mace Head is restricted to times when air masses approach from the clean air sector and the data therefore represent a baseline case for the time series of halogenated compounds. In contrast, weekly air sample collection at Taunus Observatory is performed irrespective of wind direction. Therefore mixing ratios at Taunus Observatory are supposed to be higher than at the coastal site except for substances which are strongly influenced by marine emissions, such as for example carbonyl sulfide or iodomethane (not shown here). Because Taunus Observatory is located closer to sources, not only higher absolute mixing ratios of most halocarbons are expected to be measured but also a higher atmospheric variability for substances with ongoing emissions.





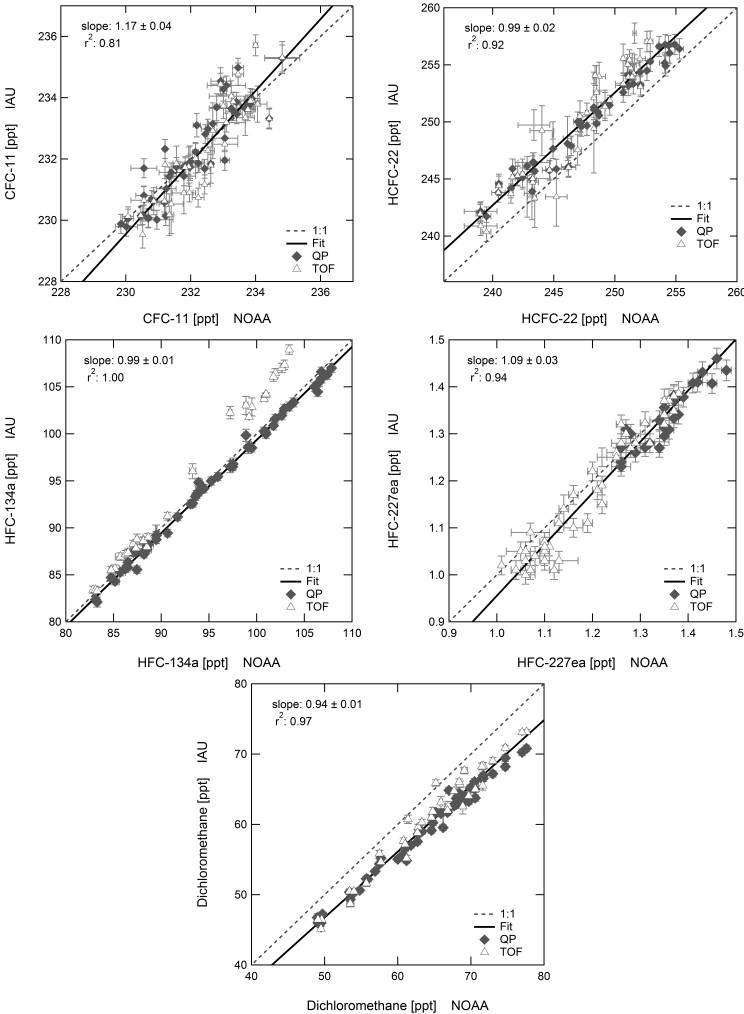

**Figure 3.** Correlation of CFC-11, HCFC-22, HFC-134a, HFC-227ea, and dichloromethane in Frankfurt flask samples (IAU) with canister samples of the NOAA network analysed by GC/MS (data available at: ftp://ftp.cmdl.noaa.gov/hats/) at Mace Head. Results from the Frankfurt GC/MS system are reported on AGAGE scales resulting in a constant offset for some substances. Because of the known non-linearity of the TOF instrument, for HFC-134a the correlation parameters are derived for the QP mass spectrometer only.

**(Hydro)Chlorofluorocarbons: CFC-11 and CFC-12 and HCFC-22**

The Montreal Protocol strictly regulates production and use of CFC-11 and CFC-12. Due to their long total atmospheric lifetimes of 52 years (CFC-11) and 102 years (CFC-12) (SPARC, 2013), tropospheric mixing ratios decrease slowly but continuously, which is evident at Taunus Observatory and at Mace Head as shown in Fig. 2. Remarkable in the time series is





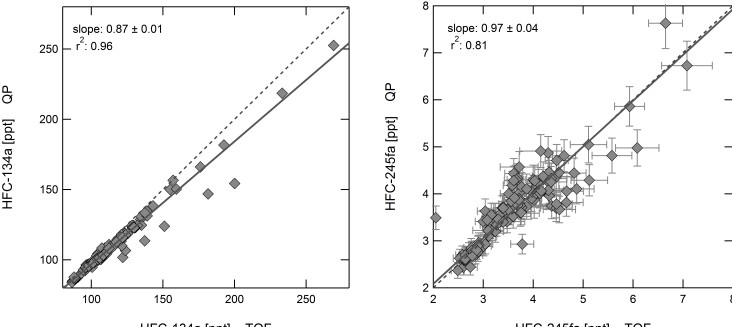

**Figure 4.** Correlation at Taunus Observatory of HFC-134a and HFC-245 for the mass spectrometers employed, a quadrupole (QP) and a time-of-flight (TOF) mass spectrometer. The solid line represents an orthogonal fit to the data points, the dotted line indicates the 1:1-line. For HFC-134a a non-linearity is observed for the TOF-MS but not for HFC-245fa. Error bars of HFC-134a mixing ratios are smaller than symbol size.

**Table 2.** Atmospheric lifetimes of CFCs, HCFCs, HFCs (SPARC, 2013), and dichloromethane (Hossaini et al., 2017) and Global Warming Potentials (GWP) (IPCC, 2013) of the discussed species.

| compound | total lifetime $\tau$ [a] | $\tau_{OH}$ [a] | GWP |
|---|---|---|---|
| CFC-11 | 52 | — | 6900 |
| CFC-12 | 102 | — | 10800 |
| HCFC-22 | 12 | 10.8 | 218 |
| HFC-134a | 14 | 14.5 | 3710 |
| HFC-227ea | 36 | 45.3 | 5360 |
| HFC-245fa | 7.9 | 7.8 | 2920 |
| Dichloromethane | 0.43 | | 33 |

an episode in September 2016 with exceptionally low mixing ratios occurring at both, TO and MHD, which is apparent for CFC-11 and CFC-12, but is more pronounced for CFC-12. A CFC-12 mixing ratio of only 509.7 ppt was measured on 16. September 2016. Flasks from the two sites for this period were analysed on different days, making a measurement artefact unlikely.

5   Comparing the means of the detrended time series, TO baseline data agree with measurements at Mace Head for CFC-11 and CFC-12 within their respective standard deviation, although mixing ratios at TO are on average higher by 1 ppt (0.5 %) for CFC-11 but only 0.4 ppt (< 0.1 %) higher for CFC-12. Applying a linear fit function to the time series, baseline mixing ratios of CFC-11 at TO decrease at a rate of -1.2 ± 0.1 ppt/year, at MHD with a rate of -1.0 ± 0.1 ppt/year. This result agrees with the global decrease rate of -1.0 ± 0.2 ppt/year determined for the period 2015–2017 at NOAA background measurement

10  sites (Montzka et al., 2018).

    In addition to the expected overall similar behaviour of CFC-11 and CFC-12, also some differences become apparent in the two gases' time series at Taunus Observatory. For CFC-11, the outlier filter routine identifies four samples in the time series at MHD as outliers, two below and two above the baseline variability, the latter two occurring in winter 2017/2018. For CFC-12



no outliers above the baseline occur at MHD. At Taunus Observatory, CFC-11 also has a larger number of polluted outliers with exceptionally high mixing ratios (18 %, 28 of 156 datapoints) than CFC-12 (4 %, 7 of 174 datapoints). Outliers in one of the two compounds are only in one case an outlier in the other one. CFC-11 outliers above the baseline at TO are more likely to correspond to outliers in HCFC-22 (11 pairs), HFC-134a (19 pairs) or dichloromethane (16 pairs). This is similar to observations of correlations between CFC-11 and HCFC-22 and dichloromethane at Mauna Loa after 2012 by Montzka et al. (2018).

CFC-12 enhancements occurred mainly in 2014 and 2015, while for CFC-11 they are distributed over the full measurement period. In addition, CFC-11 outliers deviate stronger from the fitted baseline curve with a maximum enhancement of 22 ppt and an average enhancement of 3.9 ppt (1.6 %). CFC-12 outliers show an enhancement of up to 5.5 ppt with an average of 4.6 ppt (0.9 %).

Another difference between CFC-11 and CFC-12 becomes apparent in the slope of the resulting function when applying equation 1 to the baseline time series. For CFC-12, mixing ratios steadily decrease. In mathematical terms, the first derivative of the fit curve is negative at all times. This is not the case for CFC-11, for which the first derivative of the fit function periodically becomes positive, indicating short periods of increasing mixing ratios. This occurs during summer months when mixing ratios slightly increase after a spring minimum. An average increase of 0.1 ppt occurs between the minimum value of the fitted baseline curve in June and in September when the annual maximum is observed. Note that these values are derived from the baseline fit curve without detrending and should not be interpreted as average seasonal cycles. For CFC-12, values of the fitted baseline curve in September are lower than in June in all years by on average 0.7 ppt. This behaviour points to higher ongoing emissions of CFC-11 than of CFC-12 in regions that influence the observation site.

Production and use of both CFCs are regulated, and their emissions should slowly approach zero. Their seasonality should therefore be driven mainly by transport patterns, in particular the intrusion of aged air with lower mixing ratios from the lower stratosphere (Rosenlof, 1995; Škerlak et al., 2014). To assess the seasonality of mixing ratios, baseline time series were detrended relative to January 2013 by subtracting the linear and quadratic term of equation 1. Fig. 5 shows the resulting seasonal cycles for the two CFCs as differences to their respective annual mean for TO and MHD. Shown are monthly means and error bars indicate the error of the mean.

In Fig. 5, both CFCs show elevated mixing ratios in winter with regard to the annual mean and reach minimum mixing ratios in spring/summer. Commonly, such behaviour occurs for gases which are predominantly removed from the atmosphere by reaction with OH or/and have increased winter time emissions which in Europe is typically the case for combustion products. CFC-11 and CFC-12 do not have an OH sink and their emissions are not related to combustion processes. The seasonal cycle of CFCs is driven by the seasonality of stratosphere-troposphere exchange which in the northern hemisphere maximizes in late winter and spring (Škerlak et al., 2014). If remaining emission sources were regionally co-located, seasonality of large-scale transport patterns should affect CFC-11 and CFC-12 similarly. In Fig. 5, the cycle of CFC-11 is shifted forward by about two months in comparison to that of CFC-12. Another difference between the two compounds with regard to seasonality is that, as mentioned above, the fit curve of CFC-11 has a positive curvature in summer pointing to a small periodic increase in mixing ratios whereas for CFC-12 curvature is negative at all times, which means that mixing ratios continuously decrease. This holds for MHD and TO.

These observational differences between CFC-11 and CFC-12 are in agreement with higher estimates of ongoing European emissions for CFC-11 than for CFC-12 (Keller et al., 2012). Although this was derived for the year 2009, it is likely that remaining emissions from banks still show this behaviour. In particular the high number of outliers is indicative of ongoing



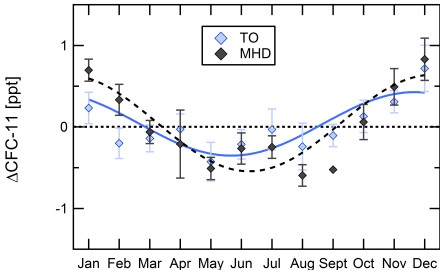
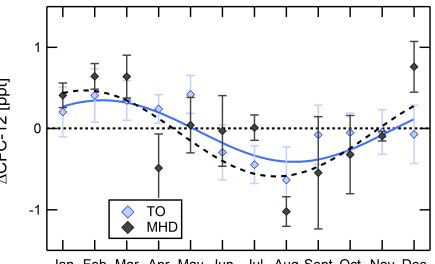

**Figure 5.** Seasonality of CFC-11 (a) and CFC-12 (b) at Taunus Observatory (coloured symbols, baseline data only) and Mace Head (black symbols). Plotted are monthly mean values as difference to the mean of the total detrended data.

emissions of CFC-11 in regions that influence trace gas mixing ratios at Taunus Observatory. New significant sources of CFC-11 were only recently reported in Asia by Montzka et al. (2018).

As an example of the first generation replacement substances, Fig. 6 shows time series and seasonal cycle of HCFC-22. HCFC-22 is widely used as a cooling agent for refrigeration and air conditioning as well as for foam blowing and for production

of synthetic polymers. Production and use of HCFC-22 have been regulated globally and are projected to be almost completely phased-out worldwide by 2030. EU regulation has banned the use of fluorinated greenhouse gases with global warming potentials higher than 150 even earlier, depending on the type of application. Globally, HCFC-22 mixing ratios are still increasing as evident from ground based measurements (Carpenter et al., 2014) and also from MIPAS satellite data (Chirkov et al., 2016). Already for the period 2005–2009 emissions had been diagnosed to at best stagnate in some regions such as parts of North

America and Europe with no significant emissions changes, while still increasing globally, mainly due to rising emissions from Asia and Africa (Saikawa et al., 2012). Graziosi et al. (2015) found European emissions to decrease for the period 2002–2012. In this Bayesian inversion study using data from Monte Cimone and from Mace Head, European emissions were estimated on the national level with emissions occurring all over Europe but predominantly in Western Europe.

The main removing process of HCFC-22 from the atmosphere is via reaction with OH with a lifetime of 10.8 years (SPARC,

2013). Thus, a seasonality of atmospheric mixing ratios with a summer minimum and a winter maximum is expected. However, for both observational sites, Taunus Observatory and Mace Head, a semi-annual cycle, adding higher harmonic terms to the fit equation yields a better fit to the seasonal cycle derived from the detrended data as shown in 6 (b) (cf. equation 2).

$$\chi(t) = a + b \cdot t + c \cdot t^2 + d \cdot \sin(2\pi t) + e \cdot \cos(2\pi t) + f \cdot \sin(4\pi t) + g \cdot \cos(4\pi t). \tag{2}$$

This has been taken into account for flagging individual samples as outliers. Of 168 valid data points, 27 are identified as

outliers above the baseline which, keeping in mind the limited statistics, occur most frequently and with highest enhancements during the summer months. The outlier frequency of the still used compound HCFC-22 is thus comparable to that of CFC-11 which should have been phased out globally and therefore should exhibit fewer outliers. HCFC-22 enhancements of up to approx. 20 ppt were measured (average 5.9 ppt (2.3 %)), CFC-11 enhancements even reached 22 ppt (average 3.9 ppt (1.6 %)). 11 outliers occurred simultaneously in HCFC-22 and in CFC-11, while only one outlier sample was found to be enhanced in

CFC-11 and CFC-12 and only four in CFC-12 and HCFC-22.





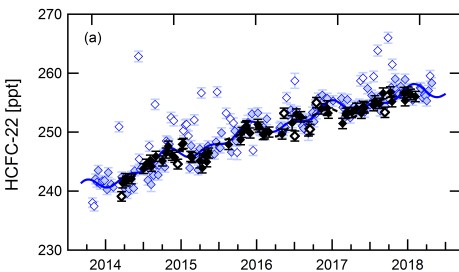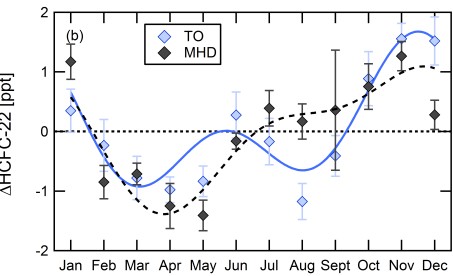

**Figure 6.** Time series (a) and seasonality (b) of HCFC-22 at Taunus Observatory (blue) and at Mace Head (black). Open symbols in the time series indicate samples flagged as outliers. Seasonality is shown as difference of the monthly mean to the mean of the full detrended baseline time series.

Graziosi et al. (2015) derived average annual growth rates of 6.9 and 7.0 ppt/year from high frequency measurements of HCFC-22 at Monte Cimone and at Mace Head for the time period 2002–2012. Growth rates increased until 2008 and started to decline afterwards with values around 3 ppt/year in 2012. At Taunus Observatory, where measurements started in late 2013, atmospheric mixing ratios steadily increase over the measurement period, with the increase rate at TO slowing down from

around 5 ppt/year in 2014 to around 3 ppt/year in 2017.

In a global inversion study using ground station data, Fortems-Cheiney et al. (2013) derived a seasonality of regional emissions of HCFC-22 possibly arising from a seasonality in the use of air conditioning and refrigeration devices. This was most pronounced in Eastern Asia, the US and the Middle East. In an inversion study based on data from NOAA and AGAGE station networks and additionally constrained by data from airborne measurements from the HIaper-Pole-to-Pole Observations

(HIPPO) missions, Xiang et al. (2014) could reproduce the observed seasonalities in HCFC-22 mixing ratios only with a seasonal adjustment to emissions with higher summertime emissions. Observations at Taunus Observatory are consistent with the assumption of seasonally varying emissions.

**Long-lived hydrofluorocarbons: HFC-134a, HFC-245fa, and HFC-227ea**

As replacement for the ozone-depleting CFCs and HCFCs, hydrofluorocarbons are now commonly used. Not containing chlo-

rine or bromine atoms, they are not ozone depleting substances. However, long-lived HFCs are strong greenhouse gases contributing to global warming (cf. Table 2). To reduce the adverse contribution of HFCs to future global warming, regulation of their use and production has been added to the Montreal Protocol in 2016 in the Kigali amendment. Several HFCs are measured regularly at AGAGE and NOAA sites and also at Taunus Observatory.

Fig. 7(a) shows the time series of HFC-245fa measured with the QP mass spectrometer. The QP dataset is used because of

better data coverage although the TOF instrument yields a better precision (cf. table 1). For the time period covered by the QP as well as the TOF mass spectrometer, the correlation of the two datasets yields a slope of $0.97 \pm 0.04$ ppt/ppt (axis offset of $0.14 \pm 0.14$ ppt) taking into account the precision of both instruments in an orthogonal data fitting routine (cf. Fig 4).

Vollmer et al. (2006) reported first measurements of HFC-245fa ($CHF_2CH_2CF_3$) in the atmosphere from Jungfraujoch (Switzerland) where mixing ratios reached 0.68 ppt at the end of 2005. At Taunus Observatory, baseline mixing ratios were

around 2.5 ppt in late 2013 and climbed up to values around 4.2 ppt by the end of 2017 after a slowdown of the increase rate



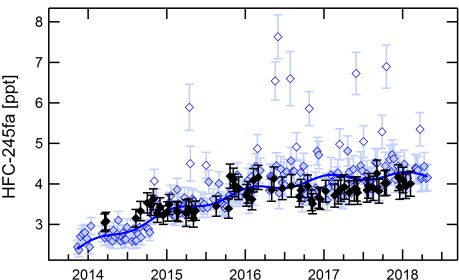 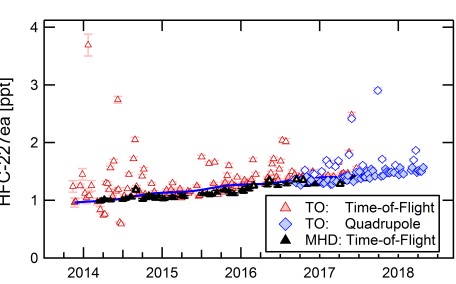

**Figure 7.** Time series of HFC-245fa (a) and of HFC-227ea (b) at Taunus Observatory (coloured symbols) and Mace Head (black symbols). HFC-227ea measurements were performed with the time-of-flight mass spectrometer (coloured diamonds), for comparison data from the quadrupole mass spectrometer (coloured triangles) are shown in addition. Solid lines represents a 2nd order polynomial harmonic fit to the TO baseline dataset. Open symbols denote samples flagged as outliers at Taunus Observatory.

from 0.8 ppt/year in 2014 to less than 0.1 ppt/year in 2017. Mace Head data overall agree with the TO baseline. Outliers were few in 2014 but their number increases through 2017. HFC-245fa exhibits a seasonal cycle with minimum mixing ratios in summer, consistent with an OH sink, and a second, less pronounced minimum in January, possibly related to transport of stratospheric air containing lower amounts of HFC-245fa. Its total lifetime is estimated to 7.9 years, dominated by loss through

reaction with OH (SPARC, 2013). Although its stratospheric lifetime is much longer, aged stratospheric air contains lower HFC-245fa mixing ratios due to its tropospheric trend.

Emissions of HFC-227ea started in the early 1990s from a range of applications, such as from fire extinguishers replacing bromine containing halons. Laube et al. (2010) first measured HFC-227ea ($CF_3CHFCF_3$) in atmospheric air samples and firn air samples. In 2009, atmospheric mixing ratios were about 0.5–0.6 ppt with higher values in the northern hemisphere. Panel

(b) of Fig. 7 displays the time series of HFC-227ea mixing ratios at Taunus Observatory and at Mace Head. HFC-227ea was only added to the quadrupole measurements in 2016. For the time period covered by both instruments, the two datasets agree with a linear correlation coefficient of $r^2 = 0.96$, the orthogonal fitting procedure yields a slope of $1.06 \pm 0.04$ ppt/ppt (axis offset of $-0.13 \pm 0.06$ ppt). This example highlights the potential of the TOF mass spectrometer for retrospective analysis of non-target compounds.

In early 2014, background mixing ratios of HFC-227ea reached approx. 1 ppt and atmospheric mixing ratios continue to increase at both sites, showing almost no seasonality as expected from its long lifetime $\tau_{OH} = 45.3$ a (SPARC, 2013). With exception of a few outliers in early 2014, mixing ratios at MHD are below TO baseline data which also exhibits a larger scatter. Both aspects are expected for a widely used compound. HFC-227ea sticks out with a large frequency of outliers, up to one third of samples contain mixing ratios significantly above the baseline variability. Outliers were particularly frequent and strong in

2014 and in 2017.

While global regulation of production and use of long-lived HFCs according to the Kigali amendment to the Montreal Protocol does not foresee a reduction prior to 2029, the states of the European Union have adopted the so-called F-gas directive (Regulation (EU) No 517/2014 on fluorinated greenhouse gases) aiming at a reduction of European emissions of fluorinated greenhouse gases by regulating their use after 2015. It is unclear if these measures already cause the observed slow down of





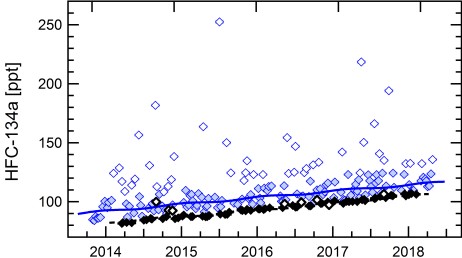

**Figure 8.** Time series (a) of HFC-134a at Taunus Observatory (blue) and at Mace Head (black). Open symbols in the time series indicate samples flagged as outliers.

HFC-245fa, but considering its lifetime with respect to reaction with OH of 7.8 years, it is a compound expected to respond to regulation rather fast. HFC-227ea with $\tau_{OH} = 45.3$ a does not show a significant slow down of the increase rate.

Additional European regulation got implemented in the mobile air conditioning directive (Directive 2006/40/EC relating to emissions from air-conditioning systems in motor vehicles). This directive restricts type approvals of vehicles fitted with an
air conditioning system operating with fluorinated greenhouse gases with a global warming potential higher than 150 after 2011. Fig. 8 shows data for HFC-134a, a widely used compound affected by this. The short-lived compound HFC-1234yf which is already used as a replacement of HFC-134a was successfully identified among the non-target species measured by our time-of-flight mass spectrometer.

Currently HFC-134a is the most prevalent HFC in the atmosphere, with baseline mixing ratios at TO varying around 120 ppt
in 2018 and a lifetime of $\tau_{OH} = 14.5$ a. Mixing ratios at Taunus Observatory are consistently approx. 10 ppt ($\approx 10\,\%$) above those at Mace Head and exhibit a large variability and a large number of outliers above the baseline. Baseline mixing ratios increase at a slightly accelerating rate of around 6–7 ppt/year. Inversion-based top-down estimates of emissions yielded increasing global emissions for the period 2004–2012 with a large discrepancy to emissions reported to UNFCCC (Rigby et al., 2014; Xiang et al., 2014).
Xiang et al. (2014) found a similar result for HFC-134a as for HCFC-22, namely that summertime emissions exceed emissions during winter. While this was reflected in the seasonal cycle of HCFC-22 mixing ratios at TO, HFC-134a baseline data at the site show only a weak seasonality due to the large variability. Because of its use in mobile air conditioning, sources of HFC-134a are ubiquitous in Central Europe. Taunus Observatory therefore is very close to emissions and the high variability in the data set masks the baseline. Mixing ratios at Mace Head are below those measured at TO throughout the observation period
and show a weak seasonality similar to that of HCFC-22 (not shown). The subset of data identified by the baseline detection algorithm does therefore not represent the European background. Still, outliers can be statistically evaluated the same way as for the other substances. The frequency of outliers observed is lowest during winter months (DJF) and has a maximum in summer (JJA). Enhancement relative to the baseline also maximises during summer. This behaviour is consistent with HFC-134a being predominantly used for air conditioning.




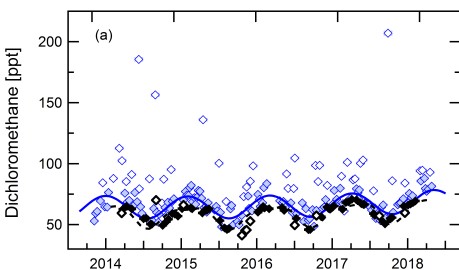
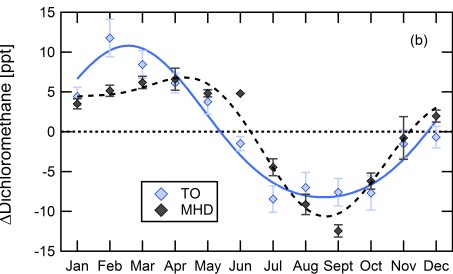

**Figure 9.** Time series (a) and seasonality (b) of dichloromethane at Taunus Observatory (blue) and at Mace Head (black). Open symbols in the time series indicate samples flagged as outliers. Seasonality is shown as difference of the monthly mean to the mean of the full detrended baseline time series.

**Dichloromethane**

As an exemplary substance with strong seasonality, Fig. 9(a) shows mixing ratios of dichloromethane. Data have been fitted using Eq. 1, as including higher order harmonics did not improve the quality of the fit. MHD data represent a lower envelope to the TO baseline time series with exception of one outlier in September 2014. The seasonal cycle is mainly driven by the

reaction of dichloromethane with the OH-radical with a lifetime of approximately 5 months (Simmonds et al., 2006; Hossaini et al., 2017). At both observation sites, seasonality shown in Fig. 9(b) exhibits an annual minimum in September and maximum mixing ratios in early spring. The distribution of outliers at TO shows the highest mixing ratio enhancements in August and September, however, most outliers occurred at other times with no season preferred.

Dichloromethane originates mainly from anthropogenic sources, it is used as a solvent and as a chemical feedstock, but

also has a minor contribution from natural sources such as oceanic emissions and biomass burning. At current, short-lived chlorinated compounds such as dichloromethane provide a small source of chlorine to the stratosphere, thus they represent a minor contribution to the stratospheric halogen load (Laube et al., 2008; Hossaini et al., 2017; Oram et al., 2017). It was recently suggested that the importance of short-lived chlorinated compounds, among them dichloromethane, as a source to the stratosphere increases as emissions in particular from Asia could rise while other contributions such as from CFCs and HCFCs

are decreasing (Hossaini et al., 2017; Oram et al., 2017).

Globally, after a period of decreasing surface mixing ratios dichloromethane levelled off around the year 2000 but started to increase again soon after. In 2013, a steep increase of surface mixing ratios occurred followed again by several years of little change (Simmonds et al., 2006; Hossaini et al., 2017). Upper tropospheric measurements over Southeast Asia revealed high spatial variability pointing to high regional emissions and rapid vertical transport (Oram et al., 2017). At Taunus Observatory,

outliers in the time series occur mainly during spring but highest enhancements in individual samples are observed in summer and autumn reaching values more than twice the respective baseline averages. On average, enhancement of outliers at TO is approx. 30 % above the baseline mixing ratio. Outliers with high mixing ratios of dichloromethane were also outliers with regard to their CFC-11 mixing ratio in 16 cases, thus in more than half of all observed CFC-11 outliers also elevated mixing ratios of dichloromethane were found.





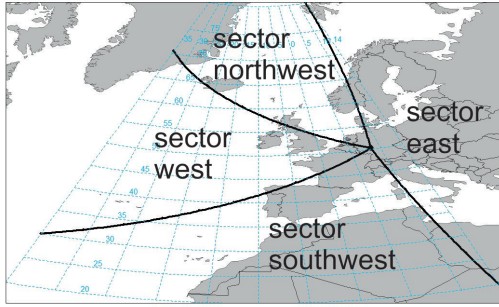

**Figure 10.** Sectoral classification of air mass trajectories. Black lines indicate sector boundaries, trajectories are attributed to a sector if more than 50 % of travel time is spent within.

In the time series shown in Fig. 9, mixing ratios were relatively stable at both sites over the period 2014–2016. From 2016 to 2017 an increase of approximately 3.5 ppt is registered at Mace Head, at Taunus Observatory, the baseline mixing ratios (annual means) increased by 2 ppt from 2016 to 2017 which is less than the annual variability of the baseline data (ca. 8.0–8.5 ppt). A more pronounced increase seems to occur at both sites in 2018.

### 3.2   Trajectory Analysis for Taunus Observatory

For a first assessment of air mass origin, HYSPLIT back trajectories were calculated over 120 h for each individual sample collected at Taunus Observatory using the 1°x 1° GDAS meteorological dataset (Stein et al., 2015). The trajectories were attributed geometrically to one of four sectors depending on their angle of approach to the site, as illustrated in Fig. 10, not taking into account altitude. A specific trajectory is counted in one sector if more than 50 % of the 120 h period is spent in it. Trajectories crossing several sectors with no sector containing more than 50 % of the trajectory points remain undefined. Trajectories of this type occur most frequently in winter (DJF). Trajectories from the west have their highest prevalence in summer (JJA) when this wind direction clearly dominates with 57 % of trajectories from that sector, in other seasons this direction contributes approx. 30 %. The number of trajectories from the easterly sector peaks with a contribution of approx. 33 % in autumn (SON) at about the same frequency as westerly air mass origin at that time of year. Winter and spring are not dominated by a particular wind direction. In all seasons, trajectories from the north-west approach TO the least often.

Fig. 11 shows the frequency with which outliers above the baseline occur in a specific sector of trajectory origin. Here, the frequency of outlier occurrence is the ratio of the number of outliers to the total number of valid samples with trajectories from a sector. Fig. 11 also shows relative mixing ratios enhancements of the samples identified as outliers relative to the baseline data. The relative enhancement of an outlier is defined as the ratio of the absolute difference between a sample's mixing ratio and the fit to the baseline evaluated at the sample collection date and the baseline fit.

Taking into account the low number of samples in each sector and the uncertainties associated with the trajectory calculation, conclusions drawn from the geometric trajectory analysis should be handled with care. The distribution of samples classified as polluted outliers at Taunus Observatory has a maximum in the south-westerly sector for all substances discussed here except for CFC-12 and HFC-245fa. Trajectories of samples with elevated CFC-12 mixing ratios most frequently point to an air mass origin from the north-west sector, which for CFC-11, HFC-134a, HFC-245fa, HFC-227ea, and for dichloromethane




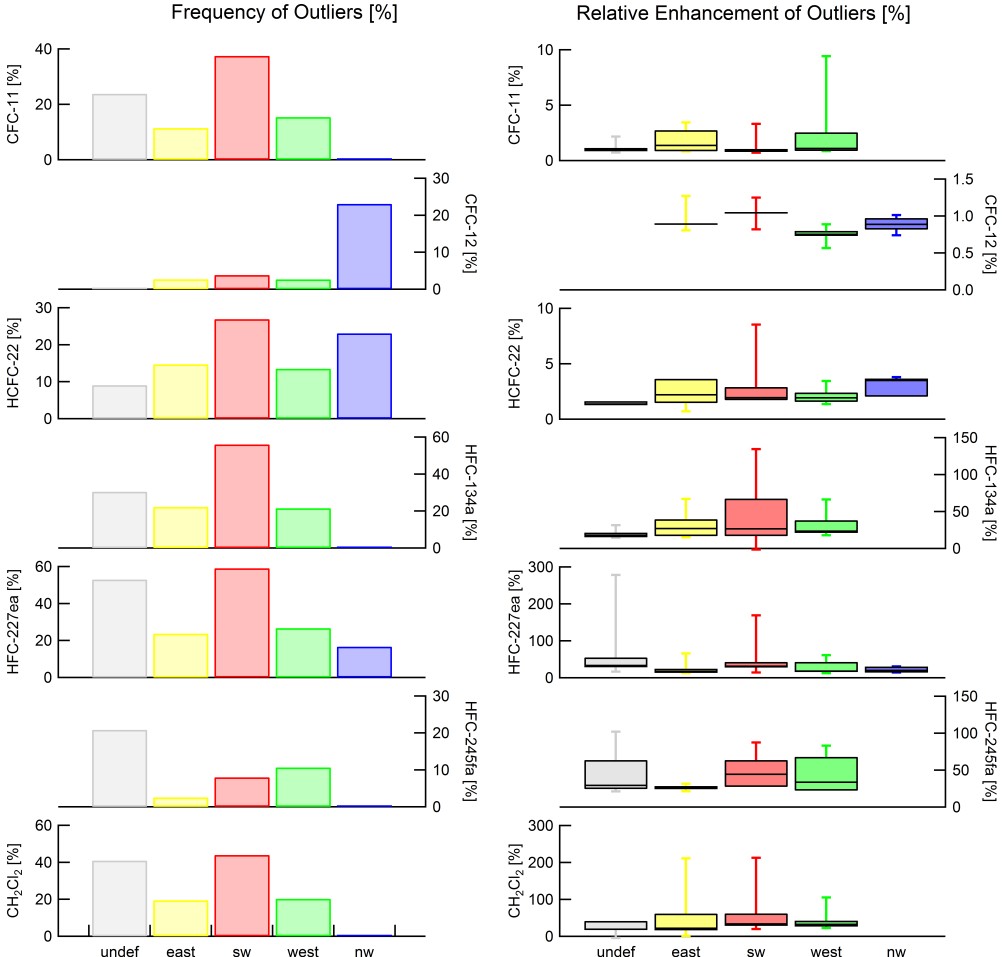

**Figure 11.** Left: Distribution of samples with enhanced mixing ratios across trajectory origin sectors. Except for CFC-12 outliers occur most frequently when trajectories originate from the south-west. Right: Box-Whisker-Plots of relative enhancement of outlier mixing ratios. Whiskers indicate minimum and maximum enhancements of mixing ratios of samples identified as outliers relative to the baseline fit. Boxes represent 25- and 75-percentiles, bars indicate the median for each sector. Remarkable features are the high relative enhancements of CFC-12 and HCFC-22 when trajectories originate from the north-west sector and high outlier enhancements of dichloromethane under easterly influence.

is the wind sector with the least occurrence of outliers. In the north-west sector fast moving trajectories approaching over the northern Atlantic at higher altitudes dominate, air masses are thus less likely to have been influenced by boundary layer pollution. Outliers of HFC-245fa occur most often in correlation with undefined trajectory origin.



Outliers in all substances except CFC-12 and HFC-245fa most frequently occur when air masses approach TO from the south-west. Air masses approaching from south westerly directions often indicate slow moving air masses which are more likely to experience surface influence. Evaluating the most distant point of each 120h-trajectory, trajectories from the south-west travel the shortest distances. Air masses of this type are likely related to the regional influence of the nearby Rhein-Main

region but might also carry emission signals from regions further south west as trajectories can reach as far as 3000 km from TO within the calculation period of 5 days.

Irrespective of trajectory sector, outliers occur most frequently with trajectories which spent the 120h period closer to TO, again for all substances presented here except CFC-12 and HFC-245fa. For these two compounds no dependency of outlier frequency with trajectory extension is apparent. Outliers associated with easterly and westerly wind directions occur at compa-

rable rates with slightly more outliers being associated with westerly winds. Trajectories from the west reach out furthest with maximum distances above 5000 km, whereas easterly trajectories are slow moving and do not extend beyond 4000 km. Outlier occurrence did not clearly correlate with any other trajectory parameters such as altitude or absolute length.

Dichloromethane has, like most of the presented substances, outliers occurring most frequently when air masses approach from the south-west, but relative enhancements can also be very high in outliers with easterly trajectories. CFC-11 exhibits

highest enhancements when air mass origin is in the westerly sector as does HFC-245fa. For the other substances discussed here, highest enhancements are associated with trajectories from the south-west with a large spread of the measured enhancement ratios.

HCFC-22, while most of its outliers are measured when air masses approach TO from the south west, also has a significant number of outliers when trajectories originate in the clean north-west sector. These samples have very high enhancements of

HCFC-22 relative to its baseline mixing ratios with the median and the 25- and 75-percentile being above those of the other sectors. The north-west sector comprises two types of trajectories, namely slowly moving ones which approach predominantly at lower altitudes and fast moving ones at higher altitudes with a higher probability of both, stratospheric and maritime impact, depending on altitude. Of these, air masses associated with a slow approach at lower altitudes might bear characteristics similar to those approaching at low pace from the west sector with a higher probability of polluted air being transported from industrial

regions in Western Germany and the Benelux countries.

## 4   Conclusions

After now more than four years of regular sample collection, we presented the first results of halocarbon measurements at Taunus Observatory for CFC-11, CFC-12, HCFC-22, HFC-134a, HFC-227ea, HFC-245fa, and for dichloromethane. Measurements are performed off-line using an automated GC/MS-system employing two mass spectrometers. Data are shown predom-

inantly from the quadrupole mass spectrometer as it yields higher data precision and has better data coverage. However, owing to the full mass scan of the time-of flight mass spectrometer operated in parallel, the number of compounds detected with this instrument is larger than for the QP instrument which is operated in SIM mode, currently detecting a pre-defined suite of 47 substances. For the time-of flight mass spectrometer, almost 60 compounds have been identified up to now, among them for example three unsaturated HFCs which are increasingly used to replace long-lived HFCs in applications such as mobile air

conditioning.

To characterise European background mixing ratios and to link the TO time series to established measurements of the AGAGE and NOAA networks, canisters collected at Mace Head Station in the site's clean wind sector are analysed with the



same setup. Mixing ratios of mainly anthropogenically influenced substances are overall lower at Mace Head than at TO, with lower variability, which reflects the vicinity to emission sources for the continental site Taunus Observatory. In addition, sampling at TO is irrespective of wind direction, while at Mace Head samples are collected when air masses approach from the clean air sector.

All data are quality filtered based on instrument precisions, and the final datasets for both sites are divided into baseline data and outliers, using an iterative outlier identification algorithm. While outliers related to pollution events with mixing ratios above the baseline variability dominate the outlier statistics, occasionally also very low mixing ratios occur.

    CFC-11 and CFC-12, for which production and use has been regulated longest, mixing ratios decrease overall, but more episodic high mixing ratio events are observed for CFC-11 than for CFC-12. Exceptionally high mixing ratios of CFC-11 most
often correlate with enhancements of HCFC-22, HFC-134a and of dichloromethane but not with enhancements of CFC-12. In addition, during summer CFC-11 mixing ratios behave different from CFC-12 mixing ratios. While the latter monotonically decrease, CFC-11 mixing ratios show a very small increase in summer following a springtime minimum.

    As an example of first-generation replacement compounds HCFC-22 is shown. The substance does not show the typical seasonal cycle expected for a compound which is predominantly removed from the atmosphere via the reaction with OH,
but exhibits a second maximum in summer. This is consistent with inversion-based model results predicting emissions of this compound widely used for cooling applications to maximise in summer. While this is also predicted for HFC-134a, almost no seasonality of this compound is observed at TO. A possible explanation is that emissions in summer dampen the seasonality imposed by reaction with OH and in addition high variability of mixing ratios masks seasonal variation. Mixing ratios of both compounds increase. This is also the case for the two other HFCs presented here, HFC-245fa and HFC-227ea, mixing ratios of
which still increase continuously at TO. However, the mixing ratio increase of the shorter-lived HFC-245fa has recently slowed down, while this is not observed for the longer-lived HFC-227ea.

    Based on a HYSPLIT trajectory analysis, most outliers are detected in air masses approaching TO from south-westerly direction. An exception to this represents CFC-12, for which the otherwise dominated by low mixing ratios north-west sector, normally associated with clean air containing background mixing ratios, has the highest occurrence of outliers above the
baseline. Also HCFC-22 outlier occurrence in this sector is very high. Maximum mixing ratio enhancements of outliers are observed when air masses arrive at the site from westerly or south-westerly directions with exception of HFC-227ea. Mixing ratio enhancements of dichloromethane can also be very high when air masses approach from the east sector.

    Halocarbon mixing ratios at TO are found to be variable with polluted outliers occurring regularly. This confirms the site's sensitivity to European emissions. Measurements of halocarbons at Taunus Observatory therefore provide an extension of
current surface data with the potential to further constrain regional European emissions, in particular as the site regularly experiences polluted conditions with air masses approaching over densely populated regions with industrial activity. Measurements will be continued and potentially extended, thus increasing the current database.

## 5   Data availability

Trace gas mixing ratio data are available from the corresponding author upon individual request.



*Acknowledgements.* The authors acknowledge the contribution of technical staff performing regular sample collection at Mace Head and at Taunus Observatory. In addition, we would like to thank D. Brunner for providing calculations for Figure 1 and S. Montzka for helpful discussion and for supplying NOAA GC/MS data.



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
