# Peer review of "Establishing Long-term Measurements of Halocarbons at Taunus Observatory"

_Atmospheric Chemistry and Physics, 2018_

## Referee Comment (RC1) · Anonymous Referee #1 · 25 Sep 2018

**Review: Establishing Long-term Measurements of Halocarbons at Taunus Observatory**

Overview:

This manuscript was an interesting read and a thoughtfully put-together paper that adds something of value to this current field of research. It outlines a newly-established regular whole air sample measurement time series based in Germany and highlights some of the current and potential uses for this dataset with several "case study" compounds. Overall I have very few suggestions to make with respect to improving the manuscript. I have outlined these below.

General comments:

- The use of "bi-weekly" is unclear as it can mean twice a week or once a fortnight (https://en.oxforddictionaries.com/definition/biweekly). Please clarify, at least at first use.
- I understand the focus on a select number of compounds for brevity but I feel that as an introductory paper more could be said in the introduction (e.g. p.3, lines 3-15) or Section 2.1 about the flask sampling programme to advertise it to others. A purpose of academic publications being the advertisement of available data for collaborations. I would suggest including (either here or in a supplement) a full list of compounds measured from the whole air samples; any ancillary measurements (e.g. pressure, temperature, wind speed) and a small description of the site: e.g. is it an existing met site with long-term measurements also taking place? If there is a website that could also be given.

Minor comments:

- P.1, line 3: I feel "allows to assess" (p.1, l3) should be "allows us to assess".
- P.3, line 28-30: does sampling always take place on the same day or at the same time? Does sampling ever occur on weekends? This may be of interest to future data users. For example there may be a difference between weekday/weekend emissions for some compounds.
- P.4, lines 2-3: please give the timeline between sample collection and analysis? What is the longest samples are stored for? What is the average storage time?
- P5, line 7: "Table 1" not "table 1".
- P5, line 9: You mention a "target standard" here but do not elaborate on this until the next page. It would be worth at least saying something along the lines of "discussed in…" here as I was left at this point thinking 'what is a target standard?'.
- Figure 2: An inset legend with visual identifies would be useful (and I believe to the ACP standard), compared to descriptions in the legend. This is also the case in Fig. 8. I also can't see a dashed line.
- P8, line 17→ and Fig. 3: Can the poorer correlation for CFC-11 be explained?
- Fig. 3: Can colours not be used as in other figures? E.g. there are colours to distinguish TOF and QP in Fig. 7.
- Table 2: I suggest this table is moved earlier in the manuscript, perhaps linked to when the compounds are introduced.
- P.11, line 15: Can increases of 0.1 ppt be determined based on the sampling frequency and analytical uncertainties?

- P.12, line 17: Do you mean "**Fig**. 6(b)"?
- I would suggest investigating other colour schemes for Fig. 11, if it needs colours at all. The green and red are not colour-blind safe and there is an intensity disparity between the yellow and grey and the blue.
- P.18: "Outliers of HFC-245fa occur most often in correlation with an undefined trajectory origin" – what is your explanation for this? Would it perhaps be better to say something along the lines of "No clear sector of origin is seen for HFC-245fa"?
- P.19, lines 1-7 (and other parts in this section): Can we say more about sources? Location of industry in these regions?
- P.19, lines 13-17: Can you provide some idea for a "why" for this section? Why does it occur most often when air comes from the this sector?
- Section 4 (Conclusions): Is there potential for this dataset to be used for emission inventory work in the future? If so perhaps touch on this.
- P.20, line 23: The sentence starting "An exception to this represents CFC-12" is rather clumsy and I recommend rewording.
- Earlier (p.17, line 21→) you mention that conclusions drawn from the trajectory analysis should be "handled with care" (due to low sample numbers and trajectory uncertainties) so I suggest repeating some of this uncertainty in the conclusion where your trajectory results seem to be stated as certain.

---

## Referee Comment (RC2) · Anonymous Referee #2 · 3 Oct 2018

This is a valuable contribution to the European and global halocarbon measurements. It is also important because it is a long-term effort, complementary to the larger networks in this field, particularly because it uses different instrumentation. The manuscript is well written and structured, perhaps a bit lengthy in some of the sections. My comments are mainly minor.

General comments: In the abstract, I suggest to list more specifically what the key findings are and avoid vague descriptions like 'differences' without stating what the main differences are.

The description and interpretation of the results, in particular p. 11–16 are rather long and could benefit from shortening to the few main important features.

[Figure]

Please publish the numerical results of the observations.

Specific comments:

p. 1, l. 2/3: I suggest to be more consistent with using the abbreviation 'TO' (introduced on line 2 but not used on line 3 and other places, or stick to the full name throughout.

While the beginning of the time series is clear (line 2), it is unclear when it ends, in particular in descriptions like p. 1, l. 15, where an understanding about the time frame is important.

p. 1, l. 3: Can you be more clear about the distinction 'local' vs 'regional'. Are local emissions really assessed here (perhaps remove the expression 'local').

p. 1, l. 8: I suggest to be more clear here in the abstract that the measurement on the two instrument is a simultaneous one using a split system. Without reading the later descriptions, this is unclear in here.

p. 1, l. 12: 'good agreement'. Can you be quantitative? Also, is the description 'with a larger variability of mixing ratios at the continental site' necessary/adding information?

p. 1, l. 14: Abbreviations like 'CFC-11', HYSPLIT etc should be spelled out the first time used both in the abstract and in the main text.

p. 1, l. 15: Rather than saying that there are 'small expected differences', could you write what these differences are? Also, I don't understand the logic of that sentence, why should there be a similar decrease in atmospheric mixing ratios, there are different types of banks, functional releases and lifetimes of CFC-11 and CFC-12.

p. 1, l. 19: 'can you be more specific than 'occurrence', perhaps mentioning frequency and/or magnitude of the pollution events over time.

p. 1, l. 20: Can you be more specific than just saying 'differences'. What are the key differences.

p. 2, l. 5: The term 'hydrochlorofluorocarbons' was introduced as the abbreviation 'HCFCs' one line above, yet here the full name is used. Check manuscript throughout for such inconsistencies. For example, the use of 'time-of-flight' and 'TOF' needs to be cleaned up also.

p. 2, l. 28: Is the factor 2.4 for Germany? Please be more specific.

p. 3, l. 6: Neither the term 'non-target' nor the 'time-of-flight mass spectrometer' have been introduced beforehand, this sentence appears to need more explanation or be removed.

p. 3, l. 6: Are the 50 compounds in addition to the selected compounds, or including them?

p. 3, l. 23: 'Additional' to what?

p. 3, l. 31: Please state the model/brand of the metal-bellows pump. Also, state what type of pump was used and how the flask samples are collected at MHD.

Figure 1: Can you make this a 2-panel figure with one of the panels zoomed in much more to see the area (e.g. 50 km radius) of the site? This would add important regional information.

p. 4, l. 6: 'ppt'. Spell out the first time used. . Also, if not done yet, specify whether this is a 'dry-air' mixing ratio or not.

p. 4, l. 10: I believe that the SI abbreviation for 'liter' is a capital 'L'. Same on next line.

p. 4, l. 20: Suggest to use 'downstream' instead of 'Behind'. Make clear, which fraction of the split refers to the TOF, and which to the Q-MS.

p. 5, l. 7: Suggest to change 'scales' to 'calibration scales' or 'primary calibration scales'. These calibration scales are SIO (Scripps Institution of Oceanography) calibration scales, and are preferably named that way, compared to 'AGAGE scales'.

[Figure]

p. 5, l. 25: Drift in what?

p. 5, l. 28/29: Be consistent with spelling 'quadrupole MS' vs 'quadrupole-MS'.

p. 5, l. 30: Suggest to change 'TOF' to 'TOF-MS'.

p. 6, Table 1 caption: 'two primary standards'. Are these the same as the two 'target standards mentioned on p. 6, l. 14. If so, I suggest to use only one of the two terms. Please clarify.

Table 1: Are the precisions 1 sigma or 2 sigma, please specify in the caption. Please change 'AGAGE scales' to SIO calibration scales (see above) and spell out SIO somewhere. Check manuscript throughout and change to SIO.

p. 6, l. 16: Not clear to me what the author means with 'slopes'.

p. 7, eq. 1: Should there be a reference to this?

p. 8, l. 9: It is not sufficient if the calibration scales are 'typically' less than 3%, there would all need to be small differences. For example, observations of dichloromethane differ about by 10% between NOAA and AGAGE and are potentially due to calibration scale differences.

p. 8, l. 11: 'high quality' is a rather subjective statement. On what grounds do the authors base this?

p. 8, l. 15: Suggest to replace 'parallel' by 'simultaneous'.

p. 8, l. 21: The implication of the sentence 'The working standard used . . ..'' is unclear to the reader. Note that the fact that the low mixing ratio in the standard does not create non-linearity per se, it would only create a factor offset. The key here and in Figure 4a is the non-unity slope between the two observational sets, the nonlinearity being created due to the large range of observations, regardless of the value for the standard. Is the nonlinearity in Fig 4a 'linear', i.e. is the solid fit line offset from 0/0 (how much)?

p. 8, l. 23: Was HFC-134a also among those substances in Hoker et al., 2015, and if so, was the nonlinearity there similar to the one mentioned in the present manuscript?

p. 10, Figure 4a: The ca 6 data points, which clearly stick out from the remaining data points, are rather puzzling, particularly given the fact that they seem to be on a single slope. Are the Q-MS data produced with EI filaments? Over the last few years, Agilent EI filaments are known for their poor behavior towards the end of their lifetime. For the duration of about 10 days before ultimate failure, under continuous use, they create a 'bimodal' response, presumable due to some shifts in the coiled filament. Note that the bimodal behavior can change quickly such that while one compound is affected in a measurement, another compound may not. Figure 4a (and perhaps Fig 4b) reminds me of that. Both NOAA and AGAGE have therefore switched to using (straight) CI filaments (still running the MSs in EI mode). While signal response is slightly reduced, signal/noise is similar, lifetime and signal drift are better for the EI filaments.

p. 10, l. 1ff: Without any interpretation, the purpose of the reporting of these low mixing ratios remains somewhat questionable. Are in-situ AGAGE data available from the internet for comparison for this time period?

p. 10, l. 5: The wording 'measurements at Mace Head' is unfortunate and confusing, the measurements were not made at Mace Head (only the AGAGE in-situ measurements were made at the site). Samples were taken at Mace Head, but measurements were done at NOAA or Uni Frankfurt.

p. 11, l. 26ff: Can you exclude seasonality in the emissions of these CFCs? It has been shown that emissions of other refrigerants are seasonally varying. For CFC-11, perhaps emissions from foam are seasonally dependent with enhanced emissions during warmer seasons?

p. 12, l. 2: Suggest to change 'in Asia' to 'for Asia'.

p. 14, l. 8: Please change 'first measured' to 'first reported'.

p. 14, l. 18: HFC-227ea is used in MDI, is there a possibility of large contamination in the lab by such device. Is lab air measured (should be mentioned in the methods). Is HFC-227ea use permitted in Germany (for applications other than MDI)?

Fig 7: There are TOF-MS HFC-227ea results far below the baseline for 2014, presumably not explainable with measurement precisions. What is the cause of this?

p. 15, l. 22: perhaps specify 'positive outliers', or 'above background' outliers.

p. 15, l. 24: AND an enhance leakage rate during that time of the year.

p. 16, l. 3: Why were higher harmonics tried? Perhaps because the fit was poor?

p. 16, l. 4: 'outlier'. A positive outlier in MHD or a negative outlier in TO? p. 16, l. 8: Is there an interpretation/explanation for this?

p. 16. In this paragraph (Dichloromethane) there is a lot of switching back and forth between observations from the data sets and published facts, making this part more difficult to read. Also, some of the facts are a repetition as they were already discussed in the introduction. Also some parts of individual sentences appear to the repetitions, e.g. the last sentence on p. 16.

p. 17, l. 4: This interpretation appears to be a bit premature given that 2018 has not finished yet.

p. 17, l. 6: Perhaps use a more convenient unit, i.e. 5 days.

p. 17. Was wind speed also used as filter?

p. 19 (trajectory analysis). Could this part be shortened? Perhaps only mention those observations, where a conclusion/interpretation is following, so that it is not heavily biased to a descriptive text.

Please provide numerical results for the compounds discussed in this paper. This is probably best done in a supplement. The data should be listed in a way that also let

the reader distinguish between background and non-background data. This could potentially be done in one single large supplementary table. Important details like which of the two instruments, which primary calibration scale and measurement precisions should be included. This is all very important for future users of these data independently on the availability through direct communication with UF.

Acknowledgments: If there are not many, why not mention the technical staff by their names.

References: Please make sure that subscripts of chemical formula are properly embedded. Please make sure that titles are consistently written with small initial letters in words other than names and the start of the title. Journal names should be abbreviated throughout.

---

## Author Comment (AC1) · 31 Oct 2018

**Establishing Long-term Measurements of Halocarbons at Taunus Observatory**
**Response to Referee #1**

We thank the referee for the thorough reading of the submitted manuscript and the helpful comments on it. While revising the draft we carefully considered the suggested modifications which are addressed point by point in the following.

*General comments:*

– *The use of "bi-weekly" is unclear as it can mean twice a week or once a fortnight (https://en.oxforddictionaries.com/definition/biweekly). Please clarify, at least at first use.*

We now use "once in two weeks" instead.

– *I understand the focus on a select number of compounds for brevity but I feel that as an introductory paper more could be said in the introduction (e.g. p.3, lines 3-15) or Section 2.1 about the flask sampling programme to advertise it to others. A purpose of academic publications being the advertisement of available data for collaborations. I would suggest including (either here or in a supplement) a full list of compounds measured from the whole air samples; any ancillary measurements (e.g. pressure, temperature, wind speed) and a small description of the site: e.g. is it an existing met site with long-term measurements also taking place? If there is a website that could also be given.*

Currently, there is no appropriate website for the station. Different activities take place at the site of which air quality monitoring and regular measurements of the German Weather Service DWD are atmosphere related, but it is new in the context of atmospheric composition measurements. Because air quality monitoring data do not meet scientific standards with regard to precision and detection limits we do not explicitly mention them in the manuscript. We have added the following information to the text: "*The site is used for different scientific and non-scientific activities including air quality monitoring and measurements by the German Weather Service.*" We will include a full list of substances which are measured from the samples in a supplementary document which is also included at the end of this reply.

*Minor comments:*

– *P.1, line 3: I feel "allows to assess" (p.1, l3) should be "allows us to assess".*

We believe the more general wording to be more appropriate as this is not restricted to the measurements we perform but would also hold for measurements of other parameters.

– *P.3, line 28-30: does sampling always take place on the same day or at the same time? Does sampling ever occur on weekends? This may be of interest to future data users. For example there may be a difference between weekday/weekend emissions for some compounds.*

Sampling takes place during weekdays only, with very few exceptions. We are aware of the fact that this introduces a bias to the dataset. Measurements at the site are currently intensified which will enable us to investigate both, diurnal variations and weekday dependence in the future. Samples are collected during daytime only, but the exact time is random. We have clarified this in the manuscript with the following statement: "*Samples are collected during daytime on a weekly basis, usually on working days, at random times and irrespective of meteorological parameters such as wind direction or wind speed.*"

– *P.4, lines 2-3: please give the timeline between sample collection and analysis? What is the longest samples are stored for? What is the average storage time?*

The average storage time for Mace Head samples is approximately two months, three months at longest, storage time for Taunus Observatory samples is on average two weeks but usually not longer than five weeks. We have included this detail in the manuscript.

− *P5, line 7: "Table 1" not "table 1".*

Corrected.

− *P5, line 9: You mention a "target standard" here but do not elaborate on this until the next page. It would be worth at least saying something along the lines of "discussed in…" here as I was left at this point thinking 'what is a target standard?'.*

A brief statement about the purpose of the target measurements and a link to subsection 2.3 have been added: *"A full measurement series also includes a blank measurement of the purified helium used as carrier gas, a vacuum blank and a measurement of a target standard, the latter being used to assess long-term stability of the setup (c. f. subsection 2.3)."*

− *Figure 2: An inset legend with visual identifies would be useful (and I believe to the ACP standard), compared to descriptions in the legend. This is also the case in Fig. 8. I also can't see a dashed line.*

Legend boxes were included in all time series figures. In addition, the fitting curves in all time series graphs were plotted over the full range of the time axes to make it better visible.

− *P8, line 17→ and Fig. 3: Can the poorer correlation for CFC-11 be explained?*

At current we do not have an explanation for this. Part of it might be attributable to the different calibrations scales, although as mentioned in the manuscript these effects are expected to be small. In a comparison of SIO and NOAA scales a small trend was found for CFC-11, though over time scales longer than covered by our data. For completeness we added the comparison for CFC-12 to Figure 3.

− *Fig. 3: Can colours not be used as in other figures? E.g. there are colours to distinguish TOF and QP in Fig. 7.*

Figure 3 shows data from Mace Head samples only which are represented by black symbols in other figures. Therefore we prefer not to use colours in Figure 3.

− *Table 2: I suggest this table is moved earlier in the manuscript, perhaps linked to when the compounds are introduced.*

The table was moved forward and is now referenced the first time at the end of the introductory section when the chosen substances are listed the first time (see page 4 of track change document).

− *P.11, line 15: Can increases of 0.1 ppt be determined based on the sampling frequency and analytical uncertainties?*

Using the quadrupole instrument, CFC-11 is measured with a precision of 0.14%. For individual data points, an average error of the mixing ratio of 0.36 is calculated. The value of 0.1 ppt refers to the fitted harmonic function and the effect occurs at both sites, TO and MHD. Because a time dependency is not considered for the parameters of the fit function, this approach averages over all years covered by the data. For an individual year the coarse sampling resolution would not allow to determine an increase of 0.1 ppt.

− *P.12, line 17: Do you mean "Fig. 6(b)"?*

Corrected.

− *I would suggest investigating other colour schemes for Fig. 11, if it needs colours at all. The green and red are not colour-blind safe and there is an intensity disparity between the yellow and grey and the blue.*

Colours are actually not absolutely needed in these graphs, therefore we think a colour-blind safe choice of colours is not as critical here as for the other figures. However, the colours facilitate the comparison between the left and right panel of the figure,

therefore we opted to stick to them. Colours were slightly modified/intensified, and the result was checked with a colour-blind simulator.

- ☐ *P.18: "Outliers of HFC-245fa occur most often in correlation with an undefined trajectory origin" – what is your explanation for this? Would it perhaps be better to say something along the lines of "No clear sector of origin is seen for HFC-245fa"?*

  A statement such as "no clear sector origin is seen" might be interpreted such that outliers are evenly distributed across the sectors of airmass origin. However, this is not the case but we see no outliers from the northwest sector, few from the east and many outliers among the samples for which trajectories cannot be attributed to a sector. Therefore we prefer to stick to the original wording.

- ☐ *P.19, lines 1-7 (and other parts in this section): Can we say more about sources? Location of industry in these regions?*

  At this stage of the data analysis and based on HYSPLIT trajectories only we consider it too speculative to write about locations of specific industrial sources.

- ☐ *P.19, lines 13-17: Can you provide some idea for a "why" for this section? Why does it occur most often when air comes from this sector?*

  Several of the discussed substances are used in air conditioning, including mobile air conditioning in cars. Emissions are therefore widespread and we would expect higher mixing ratios to be detected when air masses have passed industrial centres or densely populated regions. For CFC-11 in contrast, we know little about the spatial distribution of remaining sources. Keller et al. 2012 estimated that remaining emissions of CFC-12 in Eastern Europe are lower than in central and south western European countries because of the economical developments during the 20$^{th}$ century. This would certainly also hold for CFC-11. It is planned to use our dataset for inversion estimates of European emissions in the future and we hope to achieve a better constraint on emissions by adding the site to the existing database.

- ☐ *Section 4 (Conclusions): Is there potential for this dataset to be used for emission inventory work in the future? If so perhaps touch on this.*

  It is planned to use the dataset for inversion modelling but we would prefer not to elaborate on this beyond the current statement about the site's potential to improve constraints on European emission estimates.

- ☐ *P.20, line 23: The sentence starting "An exception to this represents CFC-12" is rather clumsy and I recommend rewording.*

  The sentence was reworded and shortened to make it more clear. It now reads: *"An exception to this is observed for CFC-12, for which the north-west sector has the highest occurrence of outliers above the baseline."*

- ☐ *Earlier (p.17, line 21→) you mention that conclusions drawn from the trajectory analysis should be "handled with care" (due to low sample numbers and trajectory uncertainties) so I suggest repeating some of this*

  A corresponding statement was added, stating *"Due to the limited statistics, the trajectory analysis does not allow conclusions about specific sources of the discussed compounds."*

**References**

Keller, Christoph A. and Hill, Matthias and Vollmer, Martin K. and Henne, Stephan and Brunner, Dominik and Reimann, Stefan and O'Doherty, Simon and Arduini, Jgor and Maione, Michela and Ferenczi, Zita and Haszpra, Laszlo and Manning, Alistair J. and Peter, Thomas; *European Emissions of Halogenated Greenhouse Gases Inferred from Atmospheric Measurements;* Environmental Science and Technology (2012); doi: 10.1021/es202453j.

**Establishing Long-term Measurements of Halocarbons at Taunus Observatory**
**Response to Referee #2**

We thank the referee for the detailed comments. Please find them addressed below

*General comments:*

- *In the abstract, I suggest to list more specifically what the key findings are and avoid vague descriptions like 'differences' without stating what the main differences are.*

   The manuscript's abstract is rather long, thus we refrain from adding more details. The mentioned differences between CFC-11 and CFC-12 are specified to be related to seasonality and outlier frequency, further details would make the abstract even more lengthy.

- *The description and interpretation of the results, in particular p. 11–16 are rather long and could benefit from shortening to the few main important features.*

   We agree that this part is long and detailed and contains several descriptive passages and we carefully though about all its subsections. However, presenting six different substances representing three groups of compounds, we would like to give similar weight to each of them. We have removed individual statements that merely described trends of the time series or are not needed fur further interpretation, such as  lines 7-11 on page 11, lines 15-18 on page 11, details in line 23, page 13 – line 2, page 14. For further modifications of this part we refer to the track change version of the manuscript.

*Specific comments:*

- *p. 1, l. 2/3: I suggest to be more consistent with using the abbreviation 'TO' (introduced on line 2 but not used on line 3 and other places, or stick to the full name throughout. While the beginning of the time series is clear (line 2), it is unclear when it ends, in particular in descriptions like p. 1, l. 15, where an understanding about the time frame is important.*

   TO data cover the time period October 2013 through April 2018, MHD data are from March 2014 through February 2018. This information has been added in section 2.1 (Sample Collection). Usage of 'TO' was reduced and the full name used instead, except for figure legends.

- *p. 1, l. 3: Can you be more clear about the distinction 'local' vs 'regional'. Are local emissions really assessed here (perhaps remove the expression 'local').*

   We agree that is was left unclear what exactly is mean by local and regional in terms of distance to the observation site. The expression 'local' was dropped because with regard to emissions of anthropogenic compounds, such as the discussed halocarbons, the site will be mostly influenced by emissions from the Rhein-Main area which would not be considered local. This would be different for biogenic emissions from the surrounding forests.

- *p. 1, l. 8: I suggest to be more clear here in the abstract that the measurement on the two instrument is a simultaneous one using a split system. Without reading the later descriptions, this is unclear in here.*

   It was added to the abstract that the quadrupole and time-of-flight mass spectrometers are operated simultaneously.

- *p. 1, l. 12: 'good agreement'. Can you be quantitative? Also, is the description 'with a larger variability of mixing ratios at the continental site' necessary/adding information?*

   Baseline data for both sites are discussed and compared in more detail in the text. Being more quantitative here would make the abstract rather lengthy. The statement on

variability we believe to be an important detail - though it states what you would expect comparing a continental to a coastal site.

- *p. 1, l. 14: Abbreviations like 'CFC-11', HYSPLIT etc should be spelled out the first time used both in the abstract and in the main text.*

    Usage of abbreviations was checked and revised when necessary.

- *p. 1, l. 15: Rather than saying that there are 'small expected differences', could you write what these differences are? Also, I don't understand the logic of that sentence, why should there be a similar decrease in atmospheric mixing ratios, there are different types of banks, functional releases and lifetimes of CFC-11 and CFC-12.*

    Overall we would expect both compounds to exhibit a similar decrease in atmospheric mixing ratios which is determined by the processes mentioned here, namely their differing lifetimes and remaining emissions from banks. However, this is not what we observe with regard to seasonality (small springtime increase for CFC-11) and outliers (more frequent outlier occurrence for CFC-11). This is described in section 3.1 but cannot be fully addressed in the abstract.

- *p. 1, l. 19: 'can you be more specific than 'occurrence', perhaps mentioning frequency and/or magnitude of the pollution events over time.*

    "Occurrence" was replaced by "frequency and relative enhancement".

- *p. 1, l. 20: Can you be more specific than just saying 'differences'. What are the key differences.*

    The key differences are frequency and relative enhancement of outliers as well as seasonality as mentioned in the next sentence.

- *p. 2, l. 5: The term 'hydrochlorofluorocarbons' was introduced as the abbreviation 'HCFCs' one line above, yet here the full name is used. Check manuscript throughout for such inconsistencies. For example, the use of 'time-of-flight' and 'TOF' needs to be cleaned up also.*

    Usage of abbreviations was checked and revised.

- *p. 2, l. 28: Is the factor 2.4 for Germany? Please be more specific.*

    *Brunner et al.* 2017 give the factor 2.4 for Germany, referring to the model median. The manuscript has in the meantime been revised and accepted for publication and the final version published in Atmospheric Chemistry and Physics is now cited rather than the discussion version.

- *p. 3, l. 6: Neither the term 'non-target' nor the 'time-of-flight mass spectrometer' have been introduced beforehand, this sentence appears to need more explanation or be removed.*

    The sentence was reworded to *"The measurements include a large suite of more than 40 known target species of chlorine-, bromine- and iodine-containing gases measured at preselected mass windows with a quadrupole mass spectrometer. In addition, non-target information of the full mass range is available from a time-of-flight mass spectrometer. More than 50 compounds have been identified in the mass spectra from this instrument. A full list of substances which were identified in the chromatograms and for which calibration data is available is included in the supplements."*

- *p. 3, l. 6: Are the 50 compounds in addition to the selected compounds, or including them?*

    The wording *"More than 50 compounds"* includes the compounds presented here. Following a suggestion by referee#2 the full list of compounds is included with the revised version of the manuscript as a supplementary document and at the end of this reply.

- *p. 3, l. 23: 'Additional' to what?*

    'Additional' with regard to data from the existing station network.

- *p. 3, l. 31: Please state the model/brand of the metal-bellows pump. Also, state what type of pump was used and how the flask samples are collected at MHD.*

    Done.

- *Figure 1: Can you make this a 2-panel figure with one of the panels zoomed in much more to see the area (e.g. 50 km radius) of the site? This would add important regional information.*

Figure 1 got replaced by a more zoomed in version but remained a one-panel figure.

- *p. 4, l. 6: 'ppt'. Spell out the first time used. . Also, if not done yet, specify whether this is a 'dry-air' mixing ratio or not.*

  Done.

- *p. 4, l. 10: I believe that the SI abbreviation for 'liter' is a capital 'L'. Same on next line.*

  Changed.

- *p. 4, l. 20: Suggest to use 'downstream' instead of 'Behind'. Make clear, which fraction of the split refers to the TOF, and which to the Q-MS.*

  We prefer to keep the current wording, but have specified more clearly how the flow is split, namely approx. 60% into the quadrupole, 40% into the time-of-flight mass spectrometer.

- *p. 5, l. 7: Suggest to change 'scales' to 'calibration scales' or 'primary calibration scales'. These calibration scales are SIO (Scripps Institution of Oceanography) calibration scales, and are preferably named that way, compared to 'AGAGE scales'.*

  'AGAGE scales' was replaced by 'SIO scales' on all occurrences.

- *p. 5, l. 25: Drift in what?*

  The term 'drift correction' in line 25 refers to detector drift during a measurement series as mentioned in lines 14/15 on the same page

- *p. 5, l. 28/29: Be consistent with spelling 'quadrupole MS' vs 'quadrupole-MS'.*
- *p. 5, l. 30: Suggest to change 'TOF' to 'TOF-MS'.*

  Usage of quadrupole MS / quadrupole-MS, TOF / TOF-MS / TOF MS was harmonized.

- *p. 6, Table 1 caption: 'two primary standards'. Are these the same as the two 'target standards mentioned on p. 6, l. 14. If so, I suggest to use only one of the two terms. Please clarify.*

  The table caption was changed to "two primary standards used as targets" because the concept of the target standard measurement to assess long-term stability does not imply that this target is a primary standard at the same time.

- *Table 1: Are the precisions 1 sigma or 2 sigma, please specify in the caption. Please change 'AGAGE scales' to SIO calibration scales (see above) and spell out SIO somewhere. Check manuscript throughout and change to SIO.*

  Precisions are 1σ ; this was included in the table caption and in the corresponding part of the text. 'AGAGE scales' was replaced by 'SIO scales' on all occurrences.

- *p. 6, l. 16: Not clear to me what the author means with 'slopes'.*

  Slopes here refers to a linear trend fitted to the time series of target measurements. The sentence was reworded to: *"Fitting a linear function to the obtained target time series, slopes agree with 0 confirming no relative drift of the primary and the working standards."*

- *p. 7, eq. 1: Should there be a reference to this?*

  In our opinion equation 1 is a general approach of fitting time series of fairly steadily increasing or decreasing compounds with a regular seasonal cycle superimposed. Although our algorithm in general follows the procedure outlined by O'Doherty et al., 2009 and similar publications, the exact equation used by O'Doherty et al. differs from our eq. 1 as it uses Legendre polynomials. Thus, there is no reference for this here.

- *p. 8, l. 9: It is not sufficient if the calibration scales are 'typically' less than 3%, there would all need to be small differences. For example, observations of dichloromethane differ about by 10% between NOAA and AGAGE and are potentially due to calibration scale differences.*

  The sentence was reworded to *'data have not been corrected for scale differences. These are for the substances discussed here less than 3 % with exception of dichloromethane (Hall et al. 2014, Carpenter et al. 2014)'*

- *p. 8, l. 11: 'high quality' is a rather subjective statement. On what grounds do the authors base this?*

  The expression 'high quality dataset' is used to differentiate between the raw data as measured and the precision filtered dataset from which outliers originating from measurement artefacts or from an unusually strong sensitivity drift during a measurement series are removed.

- *p. 8, l. 15: Suggest to replace 'parallel' by 'simultaneous'.*

  Done.

- *p. 8, l. 21: The implication of the sentence 'The working standard used . . ..." is unclear to the reader. Note that the fact that the low mixing ratio in the standard does not create non-linearity per se, it would only create a factor offset. The key here and in Figure 4a is the non-unity slope between the two observational sets, the nonlinearity being created due to the large range of observations, regardless of the value for the standard. Is the nonlinearity in Fig 4a 'linear', i.e. is the solid fit line offset from 0/0 (how much)?*

  The effect of the low mixing ratio of the standard is actually the same as the large range of mixing ratios covered by the data. In addition, if the standard is far off the atmospheric mixing ratios or the comparison standard, memory effects might occur. The line fit represented by the solid line in Fig. 4a has an axis offset of 9.4 ± 0.4 (1sigma) and thus does not agree with 0/0. This improves to an offset of 5.6 ± 0.4 (1sigma) when excluding the 6 data points for which the QP instrument yields values below the correlation (or the TOF instruments data are too high). However, there is no indication of experimental issues of one of the two instruments.

- *p. 8, l. 23: Was HFC-134a also among those substances in Hoker et al., 2015, and if so, was the nonlinearity there similar to the one mentioned in the present manuscript?*

  *Hoker et al.* 2015 discussed the non-linearity of the TOF-MS exemplary for CFC-11 (in comparison to CFC-12 for which the instrument was shown to operate linearly). HFC-134a was included in the measurement series performed then, but it was not shown in *Hoker et al.* 2015.

- *p. 10, Figure 4a: The ca 6 data points, which clearly stick out from the remaining data points, are rather puzzling, particularly given the fact that they seem to be on a single slope. Are the Q-MS data produced with EI filaments? Over the last few years, Agilent EI filaments are known for their poor behavior towards the end of their lifetime. For the duration of about 10 days before ultimate failure, under continuous use, they create a 'bimodal' response, presumable due to some shifts in the coiled filament. Note that the bimodal behavior can change quickly such that while one compound is affected in a measurement, another compound may not. Figure 4a (and perhaps Fig 4b) reminds me of that. Both NOAA and AGAGE have therefore switched to using (straight) CI filaments (still running the MSs in EI mode). While signal response is slightly reduced, signal/noise is similar, lifetime and signal drift are better for the EI filaments.*

  QP-MS data are measured with Agilent EI filaments. We do use CI filaments in another setup (usually in CI mode, though) and found them to be of varying quality with individual filaments performing poorly throughout their lifetime. For the period in question when these 6 data points were measured (June/July 2016) we did not notice any peculiar behavior of the EI filament. The filament was then in use without data sticking out in figure 4a or 4b for several months until March 2017 when it ultimately failed.

- *p. 10, l. 1ff: Without any interpretation, the purpose of the reporting of these low mixing ratios remains somewhat questionable. Are in-situ AGAGE data available from the internet for comparison for this time period?*

  These low mixing ratios are remarkable because they occur at MHD and at TO. Samples were analysed on different days and data were quality filtered. GC-MS data from AGAGE are available through 2017 ([http://agage.eas.gatech.edu/data_archive/agage/gc-ms-medusa/complete/macehead/](http://agage.eas.gatech.edu/data_archive/agage/gc-ms-medusa/complete/macehead/)). A detailed comparison of our data set with AGAGE data

from will be performed. Unfortunately, no AGAGE data for MHD seem to available for the questionable period (September 2016).

- *p. 10, l. 5: The wording 'measurements at Mace Head' is unfortunate and confusing, the measurements were not made at Mace Head (only the AGAGE in-situ measurements were made at the site). Samples were taken at Mace Head, but measurements were done at NOAA or Uni Frankfurt.*

  'TO baseline data agree with measurements at MHD' was changed to *'baseline data from the two sites (…) agree'*.

- *p. 11, l. 26ff: Can you exclude seasonality in the emissions of these CFCs? It has been shown that emissions of other refrigerants are seasonally varying. For CFC-11, perhaps emissions from foam are seasonally dependent with enhanced emissions during warmer seasons?*

  We agree that seasonality of emissions cannot be excluded and a corresponding statement has been added. The modified sentence reads: *'The seasonal cycle of CFCs is driven by the seasonality of stratosphere-troposphere exchange (…) and potential seasonal variation in emissions.'*

- *p. 12, l. 2: Suggest to change 'in Asia' to 'for Asia'.*

  Changed.

- *p. 14, l. 8: Please change 'first measured' to 'first reported'.*

  Changed.

- *p. 14, l. 18: HFC-227ea is used in MDI, is there a possibility of large contamination in the lab by such device. Is lab air measured (should be mentioned in the methods). Is HFC-227ea use permitted in Germany (for applications other than MDI)?*

  We can exclude contamination by MDI in the laboratory. All parts of the GC-MS system are leak tested regularly to exclude contamination by lab air leaking into evacuated tubing. Lab air is not measured, but we will consider to do this in the future to investigate potential contaminations.

- *Fig 7: There are TOF-MS HFC-227ea results far below the baseline for 2014, presumably not explainable with measurement precisions. What is the cause of this?*

  At current there is no explanation for these outliers below the baseline. Those in 2014 do partly also occur in the TOF data for HFC-236fa and HFC-32 (both not included in the manuscript). Outliers below the baseline do occur for most substances although not as often as outliers above the baseline. They have good measurement precisions and HFCs were found to be stable in a stainless steel canister long-term storage experiment. Therefore is seems most likely that atmospheric transport is causing these negative outliers.

- *p. 15, l. 22: perhaps specify 'positive outliers', or 'above background' outliers.*

  Changed.

- *p. 15, l. 24: AND an enhance leakage rate during that time of the year.*

  Added.

- *p. 16, l. 3: Why were higher harmonics tried? Perhaps because the fit was poor?*

  Higher harmonics were routinely tried for all substances but were only used when the fit quality was improved (based on $\chi^2$ per degree of freedom).

- *p. 16, l. 4: 'outlier'. A positive outlier in MHD or a negative outlier in TO? p. 16, l. 8: Is there an interpretation/explanation for this?*

  It was added, that the sentence refers to an outlier at Mace Head: *'with exception of one outlier at Mace Head'*. There is no obvious explanation to this outlier. Sampling at Mace Head takes place when air approaches from the clean air sector (i. e. the Atlantic). However, it cannot be totally excluded that air masses have passed over land some time before, thus, individual outliers do occur in both the Frankfurt and the NOAA data set.

- *p. 16. In this paragraph (Dichloromethane) there is a lot of switching back and forth between observations from the data sets and published facts, making this part more difficult to read. Also, some of the facts are a repetition as they were already discussed in the introduction. Also some parts of individual sentences appear to the repetitions, e.g. the last sentence on p. 16.*

  The more general statements on dichloromethane (page 16, lines 9-19) were moved to the introduction (see pages 2 and 17 of the track changes document). This part of the manuscript now focuses on the measurement results and repetitive.

- *p. 17, l. 4: This interpretation appears to be a bit premature given that 2018 has not finished yet.*

  The statement was omitted.

- *p. 17, l. 6: Perhaps use a more convenient unit, i.e. 5 days.*

  Done.

- *p. 17. Was wind speed also used as filter?*

  Dependency on local wind speed was investigated but no clear conclusion were to be drawn from this. It was therefore not included in the manuscript.

- *p. 19 (trajectory analysis). Could this part be shortened? Perhaps only mention those observations, where a conclusion/interpretation is following, so that it is not heavily biased to a descriptive text.*

  This part was shortened and slightly reorganized to become less descriptive and repetitive (see pages 19-20 of the track changes document). Dependency on local wind speed was investigated but no clear conclusion were to be drawn from this. It was therefore not included in the discussion.

- *Please provide numerical results for the compounds discussed in this paper. This is probably best done in a supplement. The data should be listed in a way that also let the reader distinguish between background and non-background data. This could potentially be done in one single large supplementary table. Important details like which of the two instruments, which primary calibration scale and measurement precisions should be included. This is all very important for future users of these data independently on the availability through direct communication with UF.*

  In general we agree with the reviewer. However, the manuscript presents a limited part of the full data set. At this stage not all compounds have been fully analyzed over the complete timeseries and not all compounds are included in the routine evaluation procedures. The station is not yet fully established and the data presented here cover a rather short period of time. To make sure that future modifications and extensions to the dataset are considered, we would like to be in close communication with data users. In addition, we would prefer future data users to work with the most recent version of the data which will be extended with time. Long-term, publication of the data set in an easier accessible format is foreseen.

- *Acknowledgments: If there are not many, why not mention the technical staff by their names.*

  Done.

- *References: Please make sure that subscripts of chemical formula are properly embedded. Please make sure that titles are consistently written with small initial letters in words other than names and the start of the title. Journal names should be abbreviated throughout.*

  Typesetting of chemical formulae was checked and corrected when necessary. Journal names were harmonized. Formatting of references will be carefully checked again during the final typesetting/proofreading process.

**References:**

[revised manuscript text omitted]

**List of substances measured with the two mass spectrometers and calibration status (as in October 2018)**

| Substance | | measured with Quadrupole-MS | identified in TOF-MS spectra | calibrated |
|---|---|---|---|---|
| 111-Trichloroethane | CH3CCl3 | x | x | x |
| 3-Chloropentafluoropropene | CF2CFCF2Cl | -- | x | -- |
| Bromochloromethane | CH2BrCl | x | x | x |
| Bromodichloromethane | CHBrCl2 | x | x | -- |
| Bromomethane | CH3Br | x | x | x |
| Carbonyl-sulfide | COS | x | x | x |
| CFC-11 | CFCl3 | x | x | x |
| CFC-112 | CFCl2CFCl2 | x | x | -- |
| CFC-113 | CCl2FCClF2 | x | x | x |
| CFC-114 | CClF2CClF2 | x | x | x |
| CFC-115 | CClF2CF3 | x | x | x |
| CFC-12 | CF2Cl2 | x | x | x |
| Chloroethane | C2H5Cl | x | x | -- |
| Chloromethane | CH3Cl | x | x | x |
| CTFE | C2F3Cl | x | x | -- |
| Dibromochloromethane | CHBr2Cl | x | x | x |
| Dibromomethane | CH2Br2 | x | x | x |
| Dichloromethane | CH2Cl2 | x | x | x |
| Halon-1202 | CF2Br2 | -- | x | -- |
| Halon-1211 | CBrClF2 | x | x | x |
| Halon-1301 | CBrF3 | x | x | x |
| Halon-2311 | CF2CHClBr | -- | x | -- |
| Halon-2402 | CBrF2CBrF2 | -- | x | x |
| HCFC-124 | CHF2CF2Cl | x | x | -- |
| HCFC-131 | CHCl2CFCl | x | x | -- |
| HCFC-132b | CF3CH2Cl | -- | x | -- |
| HCFC-133a | CF3CH2Cl | x | x | -- |
| HCFC-141b | CH3CCl2F | x | x | x |
| HCFC-142b | CH3CClF2 | x | x | x |
| HCFC-21 | CHFCl2 | -- | x | -- |
| HCFC-22 | CHClF2 | x | x | x |
| HCFC-225ca | CF2ClCF2CFCl | x | x | -- |
| HCFC-225cb | CF2ClCF2CFCl | -- | x | -- |
| HCFC-31 | CH2ClF | -- | x | -- |
| HFC-1233zd | CHClCHCF3 | -- | x | -- |
| HFC-1234yf | CH2CFCF3 | -- | x | -- |
| HFC-125 | CHF2CF3 | x | x | x |
| HFC-134a | CH2FCF3 | x | x | x |
| HFC-143a | CH3CF3 | x | x | x |
| HFC-152a | CH3CHF2 | x | x | x |
| HFC-161 | CH3CH2F | x | x | -- |
| HFC-227ea | CF3CHFCF3 | x | x | x |
| HFC-23 | CHF3 | x | x | x |
| HFC-236fa | CF3CH2CF3 | x | x | x |
| HFC-245fa | CF3CHCF2H | x | x | x |
| HFC-32 | CH2F2 | x | x | -- |
| HFC-329ccb | CF3CF2CF2CF2H | -- | x | -- |
| HFC-365mfc | CF3CH2CF2CH3 | -- | x | x |
| HFC-41 | CH3F | x | x | -- |
| HFO-1234ze | CH2CFCF3 | -- | x | -- |
| Iodomethane | CH3I | x | x | x |
| Isoflurane | CH2FCHFOCHClCF3 | -- | x | -- |
| PFC-218 | C3F8 | x | x | x |
| PFC-318 | c_C4F8 | x | x | x |
| Sulfuryl-fluoride | SO2F2 | x | x | x |
| Tetrachloroethene | C2Cl4 | x | x | x |
| Tetrachloromethane | CCl4 | x | x | x |
| Tribromomethane | CHBr3 | x | x | x |
| Trichloroethene | C2HCl3 | x | x | x |
| Trichloromethane | CHCl3 | x | x | x |